# Digital telomere measurement by long-read sequencing distinguishes healthy aging from disease

Santiago E. Sanchez[1,2,3,4,5], Yuchao Gu[1,4,5], Yan Wang[1,4,5], Anudeep Golla[1], Annika Martin [6], William Shomali[4], Dirk Hockemeyer [6,7,8], Sharon A. Savage [9] & Steven E. Artandi [1,4,5] ✉

Telomere length is an important biomarker of organismal aging and cellular replicative potential, but existing measurement methods are limited in resolution and accuracy. Here, we deploy digital telomere measurement (DTM) by nanopore sequencing to understand how distributions of human telomere length change with age and disease. We measure telomere attrition and de novo elongation with up to 30 bp resolution in genetically defined populations of human cells, in blood cells from healthy donors and in blood cells from patients with genetic defects in telomere maintenance. We find that human aging is accompanied by a progressive loss of long telomeres and an accumulation of shorter telomeres. In patients with defects in telomere maintenance, the accumulation of short telomeres is more pronounced and correlates with phenotypic severity. We apply machine learning to train a binary classification model that distinguishes healthy individuals from those with telomere biology disorders. This sequencing and bioinformatic pipeline will advance our understanding of telomere maintenance mechanisms and the use of telomere length as a clinical biomarker of aging and disease.

Telomeres are nucleoprotein structures that prevent the recognition of chromosome ends as double-strand breaks and serve to recruit telomerase, the ribonucleoprotein holoenzyme responsible for maintaining telomere length in stem cells and cancer cells[1–4]. Telomeric repeats--stretches of non-coding, TTAGGG repeats--are recognized by shelterin, the protein complex (*TIN2, TPP1, POT1, TRF1, TRF2, RAP1*) that protects telomeres and controls recruitment of telomerase to chromosome ends. In non-malignant cells lacking telomerase, telomeres shorten by several dozens of base pairs during each cell division due to the inability of DNA polymerase to fully replicate the lagging DNA strand[5,6]. After an extended period of telomere shortening, a subset of the shortest telomeres becomes dysfunctional or uncapped, signaling a DNA damage response that triggers replicative senescence, autophagy, or apoptosis[7,8]. In cells with disrupted *Rb* and *TP53*, dysfunctional telomeres precipitate telomere crisis--a process characterized by rampant chromosomal instability stemming from end-to-end fusions at chromosomal termini lacking functional telomeres[9,10]. The end-to-end fusions and chromosomal instability seen in crisis can be replicated by sabotaging the shelterin complex via deletion or destabilization of *TRF2*[2]. In self-renewing cells--such as stem, germ, and cancer cells--telomerase offsets the end-replication problem by directly elongating telomeres. Telomere maintenance by telomerase

[1]Stanford Cancer Institute, Stanford University School of Medicine, Stanford, CA, USA. [2]Cancer Biology Program, Stanford University School of Medicine, Stanford, CA, USA. [3]Medical Scientist Training Program, Stanford University, Stanford, CA, USA. [4]Department of Medicine, Stanford University School of Medicine, Stanford, CA, USA. [5]Department of Biochemistry, Stanford University School of Medicine, Stanford, CA, USA. [6]Department of Molecular and Cell Biology, University of California, Berkeley, CA, USA. [7]Chan Zuckerberg Biohub, San Francisco, CA, USA. [8]Innovative Genomics Institute, University of California, Berkeley, Berkeley, CA, USA. [9]Division of Cancer Epidemiology and Genetics, National Cancer Institute, Bethesda, MD, USA. ✉e-mail: sartandi@stanford.edu

depends on recruitment to telomeres by *TPP1* and subsequent retention at the single-stranded 3′ overhang end by *POT1*[11]. In cells with sufficiently high telomerase activity, such as embryonic stem cells and cancer cells, this process results in stable telomere lengths over time. In a minority of cancers, an alternative, homologous recombination-dependent telomere lengthening mechanism (ALT) is responsible for maintaining telomere length[12,13]. Telomeres shorten during physiological aging in most somatic tissues likely due to insufficient telomerase in the stem cell pool[14–17]. Individuals harboring genetic defects in genes regulating telomere maintenance develop telomere biology disorders (TBDs), a group of diseases characterized by aberrantly short telomeres and severe tissue defects[18–22]. Telomerase-deficient mice display an array of tissue failure phenotypes and have served as models of organismal aging and cancer[8,23]. Therefore, telomere length is an important biomarker for the replicative potential as well as the replicative history of a cell in cells with insufficient telomerase to indefinitely maintain telomere length, and, consequently, ensuring telomere maintenance by upregulation of telomerase or initiating ALT is required for any cell aspiring towards indefinite self-renewal.

## Results

### Digital telomere measurement by long-read sequencing measures intact telomeres at high-resolution

Telomere lengths have been measured by Southern blot (TRF) and fluorescence in situ hybridization (FISH) which produce coarse mean telomere lengths by measuring signals from telomeric probes hybridized to isolated genomic DNA or to telomeric chromatin, respectively. Telomeric probe hybridization has also been leveraged in assays for the measurement and quantification of the shortest telomeres following PCR amplification of chromosome ends (STELA) or adapter-ligated telomere restriction fragments (TeSLA)[21,22,24]. Recently, it has been demonstrated that long-read sequencing technologies such as PacBio HiFi and Oxford Nanopore (ONT) sequencing can be used to sequence and measure telomeres at enhanced resolution[25–28]. We developed a sequencing preparation and bioinformatic pipeline (Telometer) capable of reproducibly measuring telomeres from either whole-genome or telomere-enriched long reads (Fig. 1a). Telomere-containing reads were identified by aligning telomeric repeats to the chromosomal termini of the recently completed telomere-to-telomere human genome[29]. We defined the extent of each telomere as spanning the distance from the terminal repeat at the chromosome terminus to the final two consecutive repeats preceding the subtelomeric sequence anchoring the telomeric region to its reference chromosome[25]. The ability to measure the distribution of individual telomere lengths from whole-genome long-read sequencing data enables telomere length analysis to be generically accessible without any additional biochemical intervention prior to sequencing library preparation, but the retrieval of telomeric reads from whole-genome sequencing data is highly inefficient due to the paucity of telomere content relative to the rest of the human genome. Capturing the telomeric end with an oligo designed to complement both the telomeric 3′ overhang on one end and the ONT sequencing adapter on the other, in combination with restriction digestion of genomic DNA, resulted in enrichment for telomeric reads by several thousand-fold without significantly impacting the measurement of the telomere length distribution (Fig. 1b, c). Digital mean telomere length measured by long-read sequencing was highly correlated to existing gold standards, TRF Southern blot and flow-FISH (Fig. 1d–g). Bootstrapping analysis of our measurement results suggests the standard error of measurement by our method decays exponentially with additional telomere measurements, eventually resulting in a maximal precision of 30–40 base pairs (Fig. 1h).

To demonstrate whether digital telomere measurement (DTM) by long-read sequencing can accurately detect telomere shortening caused by defined defects in the telomerase pathway, we sequenced several genetically modified human embryonic stem cell (hESC) models of telomere dysfunction, including disease-related mutations in *TIN2*, and *PARN* or *TERT* deletion. *PARN* is a ribonuclease critical for the maturation of the human telomerase RNA component (hTR), and *TIN2* is a core shelterin subunit required for normal telomere metabolism. Inactivating mutations in *PARN* diminish telomerase RNA thus reducing telomerase levels, and heterozygous gain-of-function mutations in *TIN2* compromise shelterin function leading to telomere shortening[30–32]. Using DTM, we measured the telomere length distribution of homozygous *PARN* knockout (Fig. 1i, j), heterozygous and homozygous *TIN2* T284R mutant, or *TERT* knockout cells (Fig. 2a, d), and the corresponding wild-type parental hESCs where applicable. Telomeres were significantly shorter in heterozygous *TIN2* T284R/+ hESCs compared with WT controls and shortened further in homozygous *TIN2* T284R/T284R hESCs. In a *TERT* knockout hESC line (*CDKN2A^{-/-} TERT^{-/-}*, AAVS1:*TERT*^flox) passaged for 66, 78, 98, and 105 days post Cre recombinase-mediated telomerase inactivation, DTM revealed progressive telomere shortening (an average of 40 bp per day) and statistically significant differences in telomere length distributions at each passage time point (Fig. 2b–c). The shorter mean telomere lengths in genetically edited versus wild-type hESCs is also in broad agreement with analog measurements by TRF Southern blot in these same cells (Fig. 1j)[30].

Next, to verify that our method captures intact, full-length telomeres we set out to use DTM to observe de novo telomere addition by forcing a cell line with stable telomeres to overexpress the catalytic core of telomerase[33]. To this end, we transiently transfected HEK293T cells with a plasmid expressing an hTR encoding a variant telomeric template sequence (TSQ) or wild-type hTR in addition to TERT, or with GFP alone. We harvested genomic DNA after three days and performed telomere capture sequencing using capture oligos designed against either the canonical human telomere sequence (5′-TTAGGG-3′) or the TSQ variant sequence (5′-TTGCGG-3′). Telomere capture using the WT oligo successfully generated libraries from both experimental groups, consistent with the presence of WT telomere sequence overhangs in each sample. In contrast, attempting telomere capture with variant-targeting oligos failed to generate a successful sequencing library using genomic DNA harvested from cells that did not express the TSQ variant sequence, indicating an absence of variant telomere repeats in these samples (Supplementary Fig. 1). Transient transfection with either hTR or TSQ and TERT substantially increased in vitro telomerase activity and resulted in a 600 bp increase in the mean telomere length after 3 days in culture (Fig. 2f–i). The detection of de novo telomere elongation by expressing either wild-type or variant hTR indicates that our method measures full-length telomeres (Fig. 2g).

DTM is capable of measuring telomeres with chromosome, allele specificity provided a phase-annotated telomere-to-telomere reference assembly is available. To demonstrate this and test the reproducibility of DTM, we performed telomere capture sequencing on genomic DNA isolated from HG002 cells sourced from the Coriell Institute alongside a second independent laboratory. The summary statistics for the HG002 telomere length distributions obtained from both laboratories are in remarkable agreement (Supplementary Data 5), and the availability of an annotated diploid genome for the HG002 cell line allowed us to measure telomere length at chromosome, allele-specific resolution (Fig. 2h). We observe significant variability from the bulk telomere length distribution in a chromosome, and often allele-specific manner. For example, the mean telomere length is 6.8 kb longer on the p-arm of the paternal copy of chromosome 1 relative to the maternal copy. This demonstrates DTM will be a useful tool to further investigate chromosome-specific telomere length maintenance, inheritance, or perturbation.

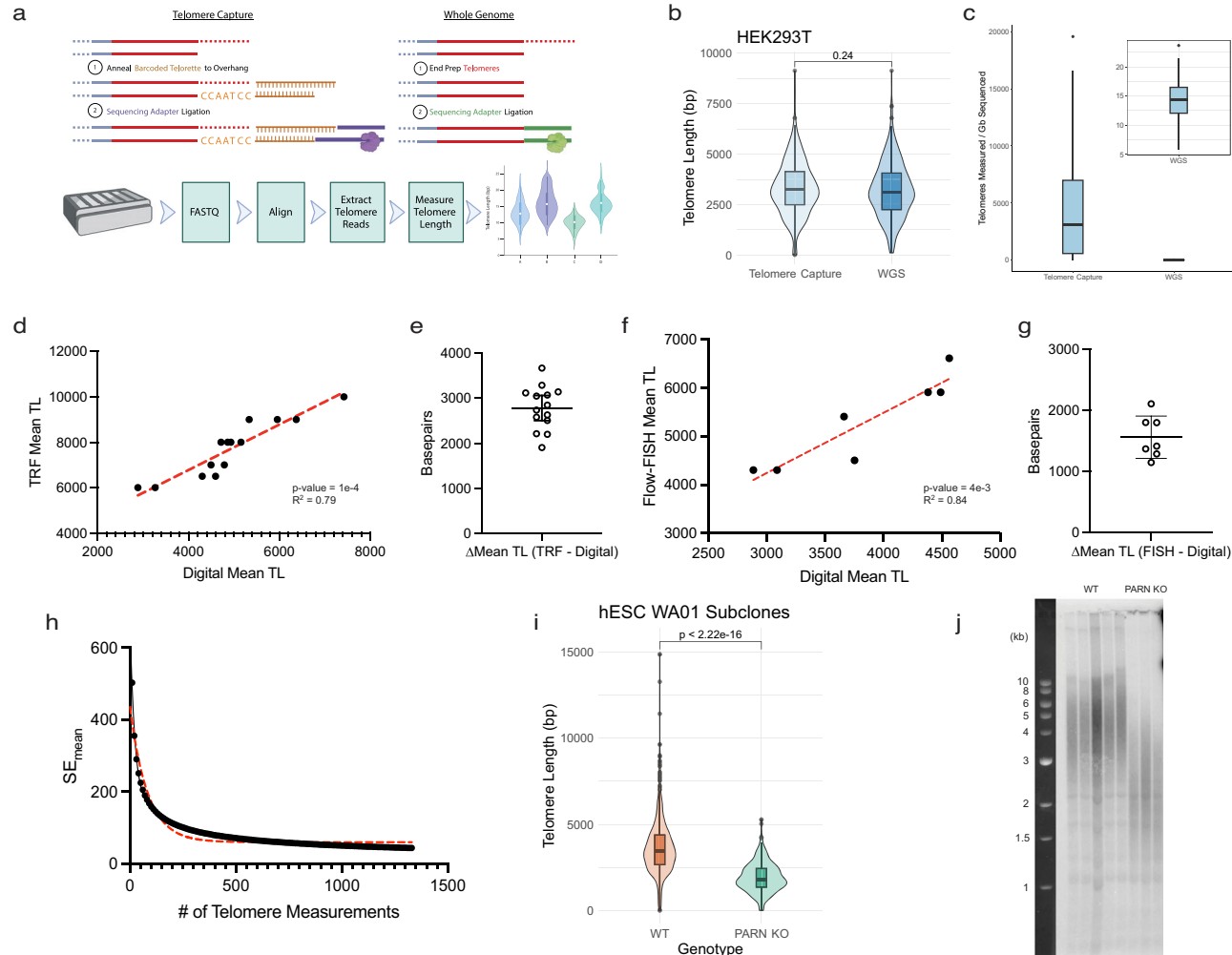

**Fig. 1 | High-resolution telomere measurement by nanopore long-read sequencing. a** Schematic representation of DNA sequencing library preparation for telomere measurement by telomere capture or whole-genome long-read sequencing. **b** Head-to-head comparison of telomere length distributions obtained through either telomere capture ($n$ = 417 telomere measurements) or whole-genome sequencing ($n$ = 513 telomere measurements) library preparation from a single source of HEK 293 T DNA. **c** Telomeres per gigabase sequenced for both library preparation methods ($n$ = 14 telomere capture experiments; $n$ = 23 WGS). **d** Correlation between mean telomere lengths from matched samples determined by sequencing or TRF ($n$ = 14, $p$ value = 1e-4, $R^2$ = 0.79). **e** The difference in bp between mean telomere length of matched samples measured by both TRF and sequencing ($n$ = 14 individual, mean ± standard deviation represented by solid lines and error bars). **f** Correlation between mean telomere lengths from matched samples of *RTEL1* mutant individuals determined by sequencing or flow-FISH ($n$ = 7, $p$ value = 4e-3, $R^2$ = 0.84). **g** Difference in bp between mean telomere length of

matched samples measured by both flow-FISH and sequencing ($n$ = 7 individuals). **h** Bootstrapping analysis of the change in standard error of the mean telomere length as a function of the total number of telomeres measured using iterative random sampling of measurements with replacement. **i** Digital telomere measurement by sequencing of wild-type ($n$ = 1333 telomeres, orange) and *PARN* KO hESCs ($n$ = 407 telomeres, green). **j** Analog telomere measurement by TRF of wild-type ($n$ = 5 independent subclones) and *PARN* KO ($n$ = 3 independent subclones) hESC subclones. For all boxplots, the 25th and 75th percentile bound the bottom and top of the box, respectively, and the central bar represents the median. Boxplot outliers are defined as having values lesser or greater than 1.5× the interquartile range (denoted by the whiskers), displayed as black dots. Two-sided Wilcoxon rank sum test was used for all quantitative comparisons. Source data provided as a source data file. **a** created with BioRender.com released under a Creative Commons Attribution-NonCommercial-NoDerivs 4.0 International license.

## High-resolution telomere measurement distinguishes human aging and disease

Mean telomere shortening in white blood cells with advancing age has been observed by previous methods of telomere measurement: Southern blot (TRF), which produces coarse estimates of the population mean; Flow-FISH and qPCR, which measures total telomeric content; and STELA or TeSLA which rely on PCR to preferentially amplify the shortest telomeres (STELA, TeSLA)[14–16,20,22]. Moreover, given the resolution of existing methods and the innate variability of telomere length between individuals of similar age, it remains unclear if telomere length can serve as a predictive biomarker of aging. We sequenced DNA derived from peripheral blood leukocytes (PBLs) from fourteen healthy human donors aged 18–77 years and found that

donor age correlated with the mean, median, first quartile, and third quartile telomere lengths (Fig. 3a–c). We observed that the mean and median telomere lengths in peripheral leukocytes decreased by approximately 27 base pairs per year, in close agreement with earlier cross-sectional estimations made from over 1100 TRFs of human PBL DNA[15]. Strikingly, the third quartile telomere length decreases more steeply with age than the first quartile (Fig. 3c), suggesting that longer telomeres are lost more rapidly than shorter telomeres and are potentially more sensitive biomarkers of aging. These data echo observations made in TRF measurements of BJ fibroblasts with limiting telomerase activity in serial passage[34] and TeSLA measurements that telomeres significantly shorter than the mean telomere length increase in proportion with age[21]. As in BJ fibroblasts, this observation could be

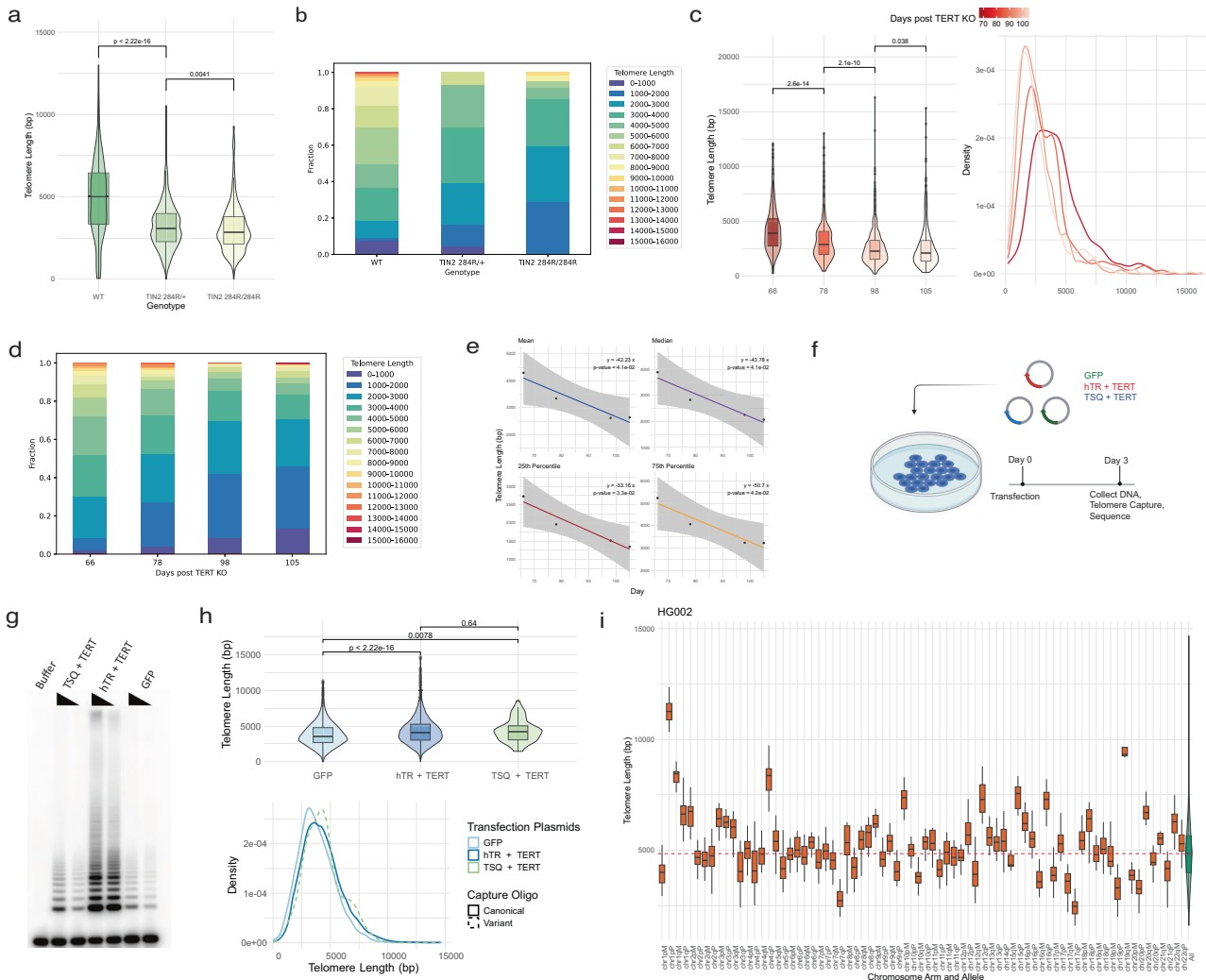

**Fig. 2 | Telomere attrition and de novo elongation in cultured human cells.**
**a** Telomere length distributions of wild-type ($n = 227$), 284 R heterozygous
($n = 1703$) or homozygous ($n = 487$) *TIN2* mutants. **b** Stacked bar graph of telomere
length fractions from *TIN2* 284 R heterozygous and homozygous mutant hESCs.
**c** Telomere length distributions of hESCs 66 ($n = 377$), 78 ($n = 350$), 98 ($n = 968$), to
108 ($n = 915$) days post Cre-mediated *TERT* knockout. Box and violin plots of telo-
mere lengths (left) alongside density distributions of telomere lengths (right).
**d** Stacked bar graphs of telomere length fractions from hESCs following *TERT*
knockout. **e** Linear regression of telomere length distribution summary statistics
versus days post *TERT* knockout ($n = 1$ for each day, 95% confidence interval in
gray). **f** Schematic representation of HEK293T telomerase overexpression experi-
ment. **g** TRAP assay for telomerase activity in transiently transfected HEK293T cells.
**h** Telomere length distributions as box or violin plots (Top) and density distribu-
tions (Bottom) for transiently transfected HEK293T cells as measured by both

canonical and variant telomere capture sequencing (GFP $n = 6424$ telomeres over 2
technical replicate runs, hTR + TERT $n = 21,402$, TSQ + TERT $n = 62$). **i** Chromosome,
allele-specific telomere length distributions (chromosome-specific boxplots in
orange; bulk distribution with violin plot for comparison in green) measured in the
HG002 cell line using a HG002 allele-annotated diploid reference genome
($n = 1862$). For all boxplots: the 25th and 75th percentile bound the bottom and top
of the box, respectively and the central bar represents the median, dotted red line
represents bulk median. Boxplot outliers are defined as having values lesser or
greater than 1.5× the interquartile range (denoted by the whiskers), displayed as
black dots. A two-sided Wilcoxon rank sum test was used for all quantitative
comparisons. Source data provided as a source data file. **f** created with BioR-
ender.com released under a Creative Commons Attribution-NonCommercial-
NoDerivs 4.0 International license.

explained by insufficient telomerase in the hematopoietic stem cell
pool to indefinitely sustain telomere length, or negative selection of
cells with increasing fractions of very short telomeres. Quantification
of telomere length fractions across our healthy aging cohort shows
that the fraction of the distribution comprised of shorter telomeres
increases with age—with the notable exception of the two shortest
fractions (0-2000 bp) which appear proportionally stable with age—as
the longer telomere fractions shrink. (Fig. 3d).

To determine if our method is capable of distinguishing healthy
aging from TBDs, we sequenced 7 samples of PBLs from 6 individuals
aged 6–50 either diagnosed with a TBD or identified as asymptomatic
heterozygous carriers of a TBD-associated *RTEL1* variant[21]. In addition,
we analyzed genomic DNA from two paired bone marrow (TB32-M)

and peripheral blood (TB32-B) samples from one patient at Stanford
Hospital with a history of chronic myelomonocytic leukemia (CMML),
pulmonary fibrosis, and a newly discovered and previously unreported
frameshift mutation in *TIN2* (p.Gln298fs) (Supplementary Data 1). We
confirmed telomeres from these individuals were significantly shorter
relative to healthy individuals in their age group (Fig. 3a, c, e) and
observed a significant correlation ($R^2 = 0.84$) with mean telomere
length measurements made by flow-FISH, the clinical standard for
telomere measurement in peripheral blood (Fig. 1f). We additionally
performed TRF with DNA from twelve of the fourteen healthy donors
and the two samples obtained from the Stanford patient (Supple-
mentary Fig. 2). TRF was found to systematically overestimate mean
telomere length by one to three thousand base pairs in all samples for

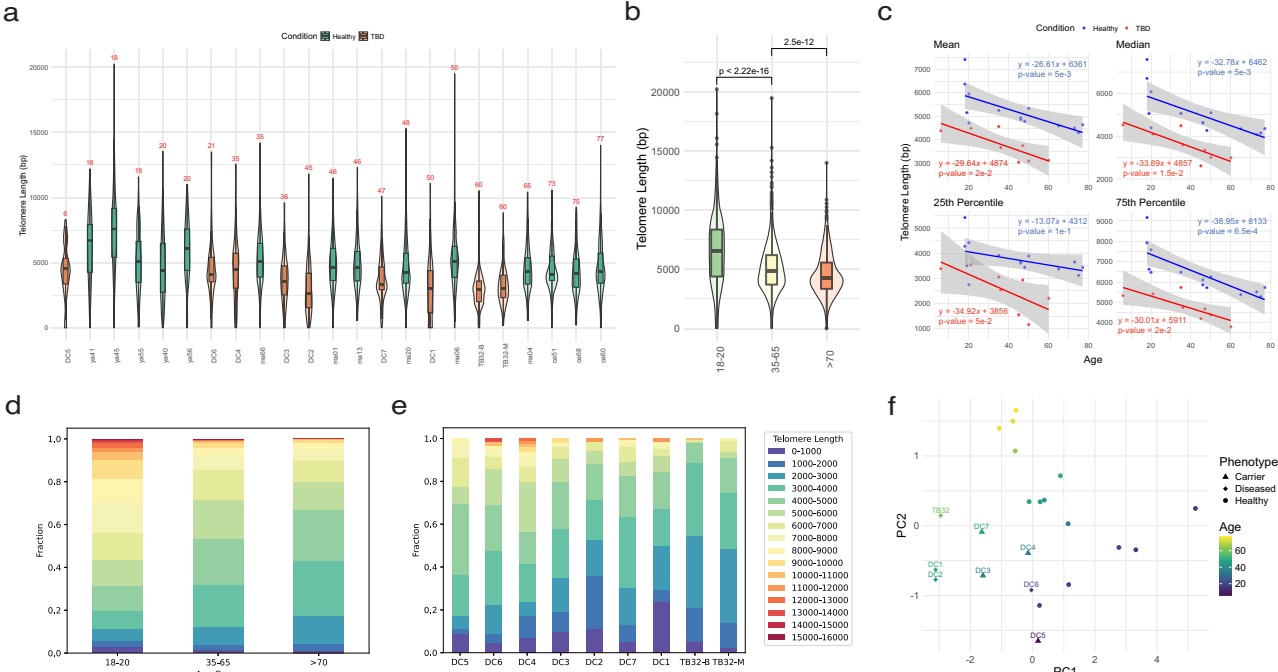

**Fig. 3 | Telomere length distributions distinguish healthy human aging and disease. a** Telomere length distributions from PBLs of 14 healthy individuals and 8 TBD variant carriers, including two samples from both PBLs (TB32-B) or a bone marrow biopsy (TB32-M) from one TBD variant carrier (precise telomere coverage per sample provided in Supplementary Data 1). **b** Telomere length distributions from PBLs of 14 healthy individuals aggregated into young (18–20; $n = 5$ individuals, $n = 834$ telomeres, green), middle (35–65; $n = 6$ individuals, $n = 1882$ telomeres, yellow), and elder (>70; $n = 3$ individuals, $n = 1036$ telomeres, orange) aged cohorts. **c** Linear regressions of telomere length summary statistics versus donor age in 14 healthy individuals (blue) or 8 TBD variant carriers (red). **d** Stacked bar graph representing telomere length fractions in 1000 bp bins from 0 to 1000 to 15,000 to 16,000 in young, middle, and elder aged cohorts. **e** Stacked bar graph representing telomere length fractions in 1000 bp bins from 0 to 1000 to 15,000 to 16,000 in 9 samples from individuals with mutations associated with, or being evaluated for, telomere biology disorders. **f** Principal component analysis of telomere length distributions of healthy aging ($n = 14$, unlabeled) and TBD variant carrier samples ($n = 9$, labeled). For all boxplots: The 25th and 75th percentile bound the bottom and top of the box, respectively and the central bar represents the median. Boxplot outliers are defined as having values lesser or greater than 1.5× the interquartile range (denoted as whiskers), displayed as black dots. A two-sided Wilcoxon rank sum test was used for all quantitative comparisons. Source data provided as a source data file.

which both digital and analog telomere measurement were performed, most likely due to undigested subtelomeric sequence biasing the results of the Southern blot (Fig. 1f, g, Supplementary Figs. 2, and 4f). Furthermore, while it was not possible to distinguish the telomere lengths from TB32's blood and marrow samples by TRF, DTM revealed the bone marrow telomere length distribution contained longer telomeres than in the same individual's PBLs (Fig. 3a, Supplementary Fig. 2). An analysis of the telomere length fractions in our cohort of individuals with defective telomere maintenance revealed significant variation between individuals but altogether much higher proportions of the shortest fractions of telomeres compared to healthy donors (Fig. 3e). Strikingly, the abundance of telomeres measuring 0–2000 bp increased in PBLs from *RTEL1* TBD patients (DC1-7) but not during healthy aging in PBLs from normal donors. The abundance of telomeres measuring 0–2000 bp also increased in *TERT* knockout hESCs with serial passage. Interestingly, engineered patient-derived *TIN2* mutations in hESCs did not lead to an increase in the abundance of short telomeres, suggesting a unique mechanism for the action of *TIN2* mutations in TBDs. In *TERT*−/− hESCs, the rate of shortening was similar for the mean, median, 25th percentile and 75th percentile telomeres (Fig. 2e). Similarly, the slope of the relationship between telomere length and age in TBD patients was comparable for all summary statistics (Fig. 3c, red lines). In contrast, during aging in PBLs from normal donors, short telomeres (25th percentile) decreased in length 3 times more gradually than long telomeres (75th percentile)(slope = −13 bp vs −35 bp, respectively)(Fig. 3c). These data suggest that the presence of intact telomerase in hematopoietic progenitor cells preferentially maintains short telomeres, blunting the rate of shortening

in normal aging. Compromising telomerase through mutations or inactivation disrupts this homeostatic mechanism, resulting in the observed similarities in the slope of the short and long telomeres in TBD patients and in *TERT*−/− hESCs. These data from both cultured stem cells and patients with dysfunctional telomere maintenance by telomerase suggest that intact telomerase function is needed to deter the accumulation of short telomeres.

Although it remains unclear what constitutes a critically short telomere, we know that the passage day 105 *TERT*−/− hESCs are 1 week away from reaching their replicative limit[32]. The abundance of short telomeres (between 0 and 2000 bp) in late passage *TERT*−/− hESCs is comparable to that observed in PBLs from TBD patients. This is evidence that bone marrow progenitors from these TBD patients are similarly approaching their replicative limit. In PBLs from normal donors, the abundance of short telomeres does not substantially increase with age (Fig. 3c). The absence of the shortest telomeres in aging PBLs suggests either that cells with very short telomeres are culled from the population or that intact telomerase actively prevents the development of these very short telomeres (0–2000 bp). These data suggest that information about the integrity of the underlying telomere maintenance machinery in a cell population can be inferred from the telomere length distribution and its change over time. Such information is not available from standard assays such as TRF Southern or Flow-FISH. Taken together, these data support a model of age-associated telomere attrition where limiting telomerase in the hematopoietic stem cell pool preferentially elongates short telomeres but is insufficient to prevent progressive shortening of the mean telomere length. This model predicts monotonic telomere shortening, on average, and more rapid shortening of longer relative to shorter

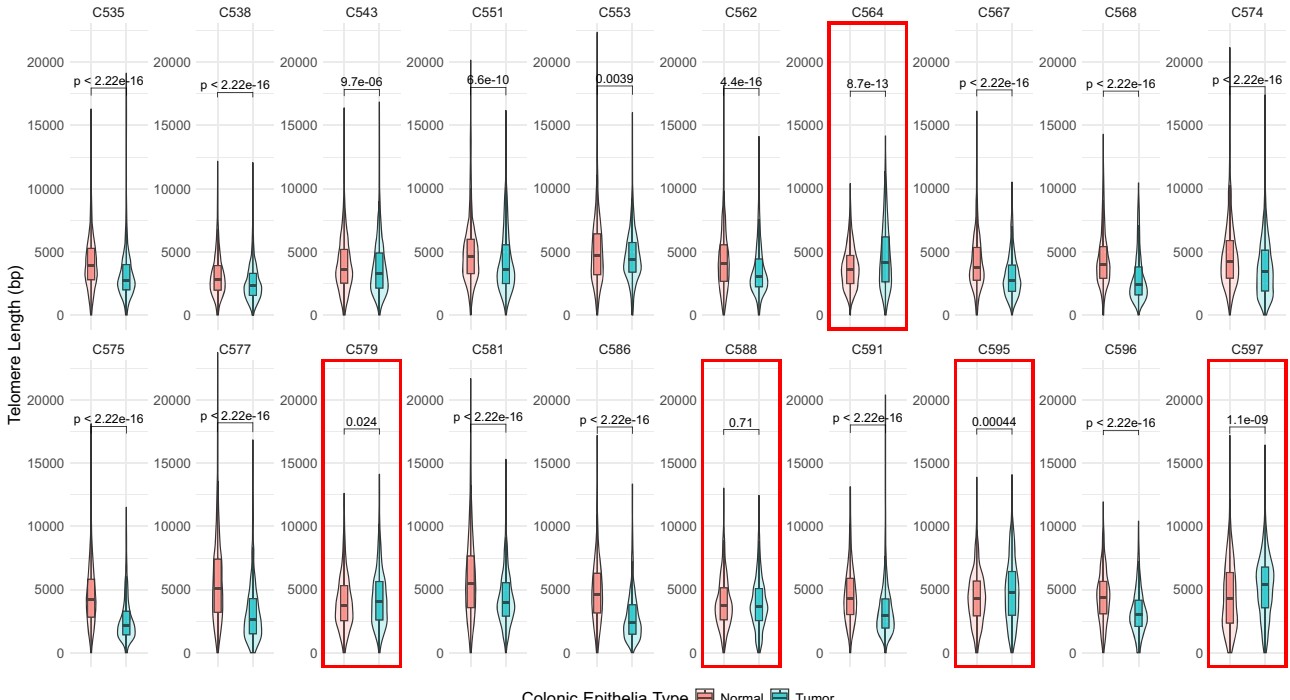

**Fig. 4 | Telomere length distributions from patient-matched benign and malignant colonic tissue in a cohort of colorectal carcinoma patients (total *n* = 37,139, precise number of telomere measurements per patient and tissue sample are in Supplementary Data 2).** Box and violin plots of telomere length distributions for twenty individuals in the cohort showing matched benign (red) and malignant (blue) colonic tissue by patient ID. Patients for whom tumors harbored telomeres equal to or longer than their benign colonic epithelia are highlighted by red boxes. For all boxplots: The 25th and 75th percentile bound the bottom and top of the box, respectively and the central bar represents the median. Boxplot outliers are defined as having values lesser or greater than 1.5× the inter-quartile range (denoted as whiskers), displayed as black dots. A two-sided Wilcoxon rank sum test was used for all quantitative comparisons. Source data is provided as a source data file.

telomeres with age in their cellular progeny and is consistent with our observations in our cross-sectional cohorts of human PBLs and genetically engineered hESCs.

TBDs are diagnosed by measuring PBL telomere length by flow-FISH, comparing the result to previously measured population statistics for telomere length for the patient's age, and subsequent detection of a genetic defect by targeted exon sequencing. Symptomatic patients often have observed mean telomere lengths at or below the first percentile for their age. TBDs also exhibit genetic anticipation, and the severity of clinical presentation is inversely proportional to telomere length. As a result, asymptomatic individuals carrying the same genetic defect as affected patients may still have telomere lengths within the lower quintile of the population for their age and subsequently develop symptoms later in life, leading to a delayed TBD diagnosis[19,21]. DTM could also be used, therefore, as a tool in the diagnosis of TBDs. To explore this possibility, we performed a principal component analysis of age and the summary statistics of the telomere length distributions from our human samples and found this information was sufficient to distinguish healthy from affected individuals (Fig. 3f). Next, we trained a logistic regression binary classification model to distinguish the following groups by telomere lengths and age: symptomatic patients versus healthy donors; unaffected carriers versus healthy donors; or both symptomatic patients and unaffected carriers versus healthy donors. Telomere measurements and ages from our healthy donors, symptomatic patients, or unaffected carriers were randomly and iteratively assigned to serve as training or test data for the binary classification model with a 50:50 training-to-test ratio, respectively. In this small sample, our model successfully distinguished both symptomatic and asymptomatic carriers from healthy donors with high sensitivity (Supplementary Fig. 5, Supplementary Data 4). In the instance of healthy donors versus symptomatic patients, our binary classification test achieved an AUC of 0.95; when comparing healthy donors versus asymptomatic carriers, an AUC of 0.90; and when comparing healthy donors versus both symptomatic patients and asymptomatic carriers, an AUC of 0.91.

Finally, to validate our observations about the relationship among age, telomere length distribution, and telomerase activity, we analyzed previously published, high-coverage (30×, on average) whole-genome long-read sequencing data from patient-matched colorectal carcinoma and surrounding benign epithelia obtained from twenty individuals predominantly aged between 50 and 70 years (Fig. 4, Supplementary Data 2)[35]. We found no correlation between age and any summary statistic of the tumor telomere length distributions, as expected for cancer cells immortalized by telomerase (Supplementary Fig. 6). However, we found that tumor telomeres were significantly shorter than those measured in matched benign epithelia in 75% of samples, in agreement with previous studies utilizing TRF, Q-FISH or TelSeq, a short-read sequencing method for estimating relative telomere content, in colorectal carcinoma, melanoma, and TCGA data, respectively[13,17,36].

## Discussion

For decades, human telomere measurement has only been possible with methods that provide either relative quantifications of telomere content or an estimation of the mean telomere length in a population. Long-read sequencing is revolutionizing our ability to explore the genomic landscape of the chromosomal terminus and we have developed a method which produces high-resolution, high-throughput measurements of individual, intact telomeres using nanopore sequencing. In this work, our method measures with high accuracy the telomere shortening observed in three TBD genotypes both in culture and in vivo, de novo telomere addition by over-expression of the catalytic core of telomerase in a cell line with stable

telomeres, and progressive telomere attrition in human blood with advancing age. We additionally report a previously undiscovered frameshift mutation in the DC-patch of *TIN2* (p.Gln298fs) in a patient with CMML, pulmonary fibrosis, and short telomeres for their age group. Telomere measurements made by long-read sequencing are comparatively much richer than TRF, flow-FISH while requiring less input DNA than either existing technique. PCR-based assays such as STELA and TeSLA can be limited by polymerase processivity and therefore reflect shorter telomeres without illuminating the entire distribution of telomeres in the sample (Supplementary Fig. 7). Our data show that TRF systematically overestimates mean telomere length by up to several thousand base pairs likely due to retention of variable sub-telomeric sequences on the terminal restriction fragment. (Supplementary Fig. 4f). Flow-FISH better estimates the mean telomere length when compared to TRF, but still overestimates telomere length by 1500 bp on average because it is calibrated to Southern blot, and flow-FISH's utility is restricted largely to mean telomere estimation in PBLs. Additionally, our observations in human aging support previous studies proposing that telomerase preferentially acts on the shortest telomeres in humans[8,34,37]. We demonstrate that it is possible to enrich a nanopore sequencing library for telomere-containing reads by several hundred-fold using a combination of custom oligonucleotides to capture telomeric ends and restriction digestion to reduce non-telomeric DNA. Nevertheless, our bioinformatic pipeline, Telometer, can be generically applied to any set of human whole-genome long-read sequencing data produced by either ONT or PacBio sequencers, but PacBio reads may underestimate telomere length due to the technology's shorter average read length (Supplementary Fig. 4).

Many studies, including this one, have demonstrated that the mean human telomere length decreases with age. In this study, we examine telomere length distributions from a cross-section of 63 healthy and diseased human samples and demonstrate the potential of DTM as a tool for clinical investigation. We make the additional observation that the structure of the telomere length distribution—interquartile telomere lengths, median, mean, fraction of telomeres of varying length—contains information about the underlying telomere maintenance mechanism of the cell. Our data suggest that the shorter end of the telomere length distribution is sensitive to the function of telomerase, and inactivating telomerase in vitro or impairing its function in vivo both have a marked impact on the maintenance of the lower end of the telomere length distribution. In the two compartments we study (peripheral blood and colonic epithelia) in aging cohorts, there appears to be some variation even in healthy aging-associated telomere attrition between the two compartments. This difference could be explained by the dosage of telomerase, the proliferation rate, or perhaps even the dosage of shelterin components within the stem cell compartments of each tissue, but a more comprehensive investigation of longitudinal telomere length evolution in healthy aging across various tissues is required. Generating ensembles of high-resolution measurements permitted us to train a proof-of-concept binary classification model capable of distinguishing healthy and diseased telomere distributions. Notably, although asymptomatic carriers are not always readily distinguishable from normal individuals by flow-FISH, we demonstrate DTM can do so with high sensitivity. Long-read sequencing technologies are already being deployed in the clinical setting for the rapid diagnosis of genetic disorders and as a tool in clinical research[38], and this work demonstrates that DTM holds promise as a diagnostic or prognostic tool for the evaluation of TBDs. As the repertoire of long-read genomic data grows, analysis of telomere length distributions from larger, more diverse, and ideally longitudinal studies will enable telomere length to become a more robust and perhaps even predictive biomarker of aging and disease.

Surprisingly, in every sequencing experiment we have performed we have measured a fraction of telomeres shorter than 1000 bp. These data may inform our understanding of what length constitutes a stable telomere, capable of binding shelterin proteins suppressing *ATM* and *ATR* responses and preventing telomere recombination activities. Future work leveraging DTM could elucidate the critical mass of very short telomeres or the thresholds at which either shelterin must be impaired or telomeres sufficiently shortened prior to triggering replicative senescence or observing telomere fusions. Access to the complete telomere length distribution of a cell and the ability to study the chromosomal, allele-specific heterogeneity of telomere length will enable telomere biologists to investigate the telomeric rheostat at much greater depth than was previously possible[39,40]. For example, in this work, we find that in a previously studied cohort of patient-matched colorectal carcinoma and benign colonic epithelia, tumor cells harbored shorter telomeres in ~75% of patients. Since telomere length distributions in cancer cells are often stable over time, it follows that the setpoint of a tumor's telomeric rheostat is somehow related to its unique cellular biology, but until now, it has not been possible to assess precisely how a certain rate of division, telomerase activity, and concentration of key telomere maintenance machinery like the RNA template, telomerase holoenzyme, or shelterin component proteins individually or quantitatively contribute to stable telomere length distributions. Understanding this dynamic biochemical equilibrium could reveal novel regulators of telomere length as well as new approaches towards treating TBDs or therapeutically targeting the telomere maintenance machinery in cancer.

## Methods

### Ethics statement
Samples from human individuals used for the preparation of this manuscript were obtained with consent and all subsequent research was conducted with all relevant ethical regulations. Study protocols were approved by IRB #13942 for the Stanford Blood Center, approving the use of genomic DNA from healthy donors for sequencing assay development, and TBD patient samples were previously collected by Sharon Savage's laboratory as a part of an ongoing NIH clinical trial (NCT00027274). The National Cancer Institute's (NCI) Institutional Review Board-approved longitudinal cohort study of inherited BMF syndromes (IBMFS) (clinicaltrials.gov NCT00027274, https://marrowfailure.cancer.gov) is a retrospective and prospective observational study. All participants or their legal guardians provide informed consent in accordance with Health and Human Services regulation 45 CFR 46. Detailed questionnaires are completed by participants, including personal medical and family history, and medical records are obtained and reviewed by medical experts.

This study was conducted in accordance with the criteria set by the Declaration of Helsinki.

### High-molecular-weight DNA isolation and quantification
High-molecular-weight (HMW) DNA was extracted using the NEB Monarch HMW DNA Extraction kit for Cells and Blood (Catalog #T3050L) according to the kit manufacturer's instructions. Briefly, for cells maintained in tissue culture, cells were trypsinized until detached and then centrifuged at $1000 \times g$ for three minutes before adding prep and lysis solution according to the manufacturer's instructions. Cells were incubated for 10 minutes at 1800 RPM in a thermomixer, and DNA from the lysed cells was precipitated onto glass beads, washed twice with 80% ethanol, and finally eluted in Monarch Elution Buffer II according to manufacturer's instructions. DNA was quantified using a Qubit 4 fluorometer and the Qubit BR dsDNA quantification reagents (Catalog #Q32850). For DNA extraction from peripheral blood, red blood cell lysis was first performed prior to DNA extraction according to the manufacturer's instructions. DNA quality was assessed by Nanodrop, and the average molecular weight was verified to be 60 kb or larger using an Agilent Tapestation (Catalog #5067-5365, 5067-5365).

## Telomere restriction fragment southern blot

Approximately 4 μg of genomic DNA was prepared in a 50 uL total volume restriction digest solution (1× Fast Digest Buffer, 3 μL HinfI, and 3 μL RsaI) and allowed to digest at 37 °C overnight. In the morning, 1 μL each of HinfI and RsaI was added to each digestion reaction and allowed to incubate at 37 °C for a further 3 hours. A 1% TAE agarose gel was prepared, and 3 μL per sample underwent gel electrophoresis (125 V, 70 minutes) to confirm restriction digest completed successfully (Fig. S2). A 0.8% TBE agarose gel was then prepared in a 20 × 27 cm casting tray after adding 15 μL ethidium bromide to the agarose solution. The entire volume of each restriction digest reaction was then loaded into each well with 1× NEB nucleic acid loading dye in addition to NEB 1 kb reference ladder, and gel electrophoresis was performed (85 V for 16 hours). In the morning, the gel was dried using a BioRad gel dryer (1 hour under vacuum then 1 hour under vacuum and 50 °C). A UV-translucent ruler was then overlayed on the dried gel before imaging with UV transillumination to establish reference distances for the ladder markers from the well positions. The dried gel was then incubated in denaturing buffer (1.5 M NaCl, 0.5 M NaOH) for one hour with gentle shaking. The denatured gel was washed with deionized water twice before a second one-hour incubation in neutralizing buffer (1.5 M NaCl, 1 M Tris-HCl, pH 7.4) with gentle shaking. The neutralizing buffer was decanted, and the neutralized gel was washed twice with deionized water. The gel was then rolled vertically into a glass hybridization tube (Thermo-Fischer Scientific) and incubated with pre-warmed hybridization buffer (Invitrogen #AM8670) at 42 °C for 30 minutes with rotation. 0.5 μM of γ-$^{32}$P labeled telomere probe was then added to the hybridization buffer tube and incubated at 42 °C overnight with rotation. The gel was washed once with 2× SSC buffer and twice more with 1× SSC buffer (0.15 M NaCl, 15 mM sodium citrate) before exposing onto a phosphor screen inside a lead exposure cassette for 24 hours. Following exposure, the phosphor screen was imaged on a Typhoon scanner.

Both the southern blot images and the ethidium bromide reference ladder were loaded onto ImageJ and aligned. The signal intensities at each position coordinate starting from the bottom of the well in the southern blot image were obtained with the ImageJ line and Measure tools after drawing a line from the bottom of the well to the bottom of the gel through each sample lane.

## Telomerase repeated amplification protocol (TRAP)

To measure telomerase activity, a two-step TRAP procedure was performed[36]. Cell protein extracts (at 1× or 3× dilution with lysis buffer per transient transfection condition) were incubated with telomeric primers for 30 min at 30 °C in a PCR machine, followed by 5 min of inactivation at 72 °C (cold extension). 1 μl of the cold extension reaction was PCR amplified (24 cycle of 30 s at 94 °C, followed by 30 s at 59 °C) in the presence of $^{32}$P end-labeled telomeric primers. The radiolabeled PCR reactions were resolved by 9% polyacrylamide gel electrophoresis at room temperature, and the gel was exposed to a phosphor-imager overnight and the phosphor screen was then scanned by a Typhoon scanner.

## Whole-genome sequencing nanopore library preparation

Nanopore library preparation for whole-genome sequencing was carried out according to Oxford Nanopore Technologies (ONT) protocol for native genomic DNA sequencing (LSK-110) with some modifications. Briefly, approximately 1 μg of DNA per sample was end-prepped using the FFPE DNA repair and Ultra II End-Prep enzyme mixes from the NEBNext companion module for ONT ligation sequencing (Catalog #E7180L). The end-prep reaction was incubated in a thermocycler at 20 °C for 30 minutes and then 65 °C for 30 minutes. End-prepped DNA was extracted from the reaction using Promega ProNex size selection beads (Catalog #NG2001) at a bead-to-reaction solution ratio of 1.6 and incubated on a Hula mixer at room temperature for 5 minutes prior to being pelleted on a magnet and then washed twice with 80% ethanol and then allowed to dry on the magnet for 3 minutes. DNA was eluted from the beads using ONT elution buffer at 37 °C for 15 minutes. For simplex experiments, sequencing adapters (ONT AMXF) were then ligated to the end-prepped DNA for one hour at room temperature using NEB Quick T4 DNA ligase (Catalog #E7180L). For multiplex experiments, ONT barcodes were ligated using NEB Blunt/TA ligase for one hour at room temperature, and barcoded DNA was extracted, pooled up to 1 μg of total DNA, and then ligated to sequencing adapters as described previously. Adapter-ligated DNA was extracted from the ligation reaction using Promega ProNex size selection beads at a bead-to-reaction solution ratio of 1.1 and incubated and eluted as described previously. 20–50 fmols of Adapter-ligated DNA was sequenced on R9.4.1 PromethION flow cells on a P2Solo for 24–72 hours. It is advisable to quantify DNA after every bead purification step using a Qubit fluorometer and the Qubit dsDNA BR DNA quantification assay.

Genetically modified hESCs were sequenced on R10.4 PromethION flow cells on a P2Solo, and therefore, the sequencing adapter used was changed to pair with the updated flow cell chemistry, per ONT's standard sequencing protocol (adapter NA instead of AMII for telomere capture).

## Telomere capture sequencing nanopore library preparation

Barcoded telomere capture oligos (Supplementary Data 3) were annealed to sequencing tether (seqTether) by mixing equimolar amounts of both oligos in low TE buffer, heating to 95 °C for 2 minutes and then being allowed to cool at room temperature for an hour. Approximately 3 μg of HMW genomic DNA was ligated to barcoded, freshly duplexed oligos in a 100 μL ligation reaction (10 μL 10× rCutSmart buffer, 5 μL 5 μM duplex capture oligos, 2 μL 2000U/μL T4 DNA ligase, 1 μL 10 mM ATP, 3 μg gDNA, nuclease-free H$_2$O up to 100 μL) overnight at 37 °C. The following day, the ligation reaction was heat-inactivated at 65 °C for 10 minutes. Potential gaps between the capture oligo and the double-strand/single-strand junction were then filled in using the same reaction tube by adding 2 μL (4U) *Sulfolubus* DNA Polymerase IV (NEB #M0327S), 12 μL 10× ThermoPol Buffer, 1 μL 20 mM dNTPs, 1 μL 10 mM ATP, and 4 μL of nuclease-free H$_2$O followed by incubation at 56 °C for 2 minutes and then 72 °C for 15 minutes with shaking (500 RPM). Promega ProNex size selection beads were then added to the ligation reaction at a bead-to-solution ratio of 1.6, and the solution was equilibrated on a rotating mixer for 5 minutes at room temperature. The bead solution was then pelleted on a magnet, washed twice with 80% ethanol, allowed to dry on the magnet for 3 minutes, and then eluted in 60 μL of ONT elution buffer for 15 minutes at 37 °C. The eluate contains the capture oligo ligated DNA, which will be referred to as the post-capture sequencing library. The complete volume of the post-capture library was then ligated onto the ONT sequencing adapter (AMII for R9, NA for R10 libraries) per ONT's barcode ligation sequencing protocol (5 μL sequencing adapter mix, 20 μL 5× Quick T4 DNA Ligase Buffer, 10 μL Quick T4 DNA Ligase, 5 μL nuclease-free H$_2$O) at room temperature for one hour. The rest of the library preparation and sequencing protocol is performed as for whole-genome sequencing above. Technical replicate comparisons of R9 and R10 flow cell chemistries and basecallers are available in the supplementary information (Supplementary Figs. 8 and 9).

To prepare multiplexed telomere capture sequencing samples, post-capture libraries containing up to 8 uniquely barcoded samples were pooled by quantifying each post-capture library with the Qubit BR dsDNA quantification assay and then combining equal molecular weights of each post-capture library up to a total of 1000 ng. Sequencing adapter ligation and subsequent preparation for flow cell loading is as previously described using the 1000 ng pool of post-capture libraries as input. The most current version of both our simplex and multiplex telomere capture methodology is available at https://github.com/santiago-es/Telometer.

## Data analysis

Sequencing data was basecalled using Guppy 6.3.0 (ONT) high-accuracy basecalling and aligned to a human telomere-to-telomere reference genome (T2T-CHM13 + Stong 2014 subtelomere assemblies[41]) with minimap2. Aligned BAM files were then sorted and indexed with samtools (v.1.16.0), and telomere measurements were extracted from alignments using a custom script. Telometer is based on previously published work for telomere measurement from PacBio HiFi reads[25], which was adapted for telomere measurement from Oxford nanopore long reads. In brief, reads mapping to the terminal arms of each chromosome are searched for telomeric repeats by using regular expressions targeted at the sequence patterns corresponding to both the canonical human telomeric repeat (TTAGGG/CCCTAA) and the commonly miscalled motifs previously identified in the literature[42]. A check is then performed to ensure that identified telomeric sequences are terminal and telomeres are measured from the read terminus until the sub/telomeric boundary. Conceptually, the sub/telomeric boundary is encountered by moving from the telomere terminus inward until telomeric repeats give way to non-telomeric sequence. We formally define the sub/telomeric boundary as the genomic position corresponding to the starting position of the final two telomeric repeats before encountering non-telomeric motifs. The telomere length is then the length, in base pairs, spanned by consecutive telomeric sequence motifs at the terminus of chromosome arms. Figures were made, and statistical analyses were performed with R (v4.1.0).

For samples sequenced on R10.4 PromethION flow cells, raw pod5 files were basecalled using a custom bonito telomere calling model (HG002.k1) provided by ONTs. As the R10 nanopore chemistry differs significantly from R9, the raw signal from the flow cell is substantially changed, and telomere identification from basecalled reads with our custom scripts is not possible with data produced from the default guppy R10 basecaller; however, ONT's custom telomere basecalling model once again makes it possible for our algorithm to accurately identify telomeric reads from R10 data. Training ONT basecalling models to basecall telomeres more accurately has been previously demonstrated in the literature[42]. Since work for this manuscript was completed, ONT has incorporated improved telomere basecalling into the default dorado basecaller (v0.3.4, dna_r10.4.1_e8.2_400bps_sup@v4.2.0). Telomere capture experiments with all three basecalling models on both sequencing chemistries are compared in Figs. S7 and S8. Besides the supplementary basecaller comparison, all experiments in the same figure were performed on the same sequencing chemistry and basecalled with the same model. Otherwise, library preparation and data analysis were performed as described above. At the time of writing, we recommend the R10 flow cell chemistry and using the high-accuracy Dorado basecaller for optimal results.

In order to measure telomeres at the level of individual chromosomes and alleles (Fig. 2h), we aligned HG002 sequencing reads to the diploid, allele-annotated HG002 reference genome, which is available on the Telometer github page (https://github.com/santiago-es/Telometer). First, one should create maternal and paternal copies of the reference genome in addition to the combined diploid genome. Reads should be aligned to both allele-specific reference genomes and analyzed with Telometer. The read id can be used to identify reads appearing in both post-analysis datasets, and only the telomere measurement corresponding to the allele with the higher mapping quality score for that read id should be preserved prior to downstream analysis.

## Binary classification model

To develop a predictor of the disease or carrier status of patients, we used LogisticRegression from Scikit-learn (v1.2.2) to train three binary classifiers on the summary statistics and age of the telomere length distributions measured in 14 healthy individuals and 8 TBD patients: healthy versus carrier, healthy versus diseased (i.e., affected carrier), and healthy versus carrier or diseased (TBD patient phenotypes are annotated in Supplementary Data 1). In training, all default values were used for LogisticRegression parameters except for max_iter (the maximum number of iterations before the solver converges), which was set to 1000. The models were trained and tested using a 50:50 trainig:test data split and a receiver-operator curve were generated with Scikit-learn, representing the variation of the true or false positive rates with the internal binary classifier threshold.

## Transient transfection of HEK293T cells

HEK293T cells were cultured in DMEM supplemented with 10% FBS and 1% penicillin/streptomycin. Transient transfection was carried out using a 3:1 polyethylenimine to plasmid ratio and 3 μg of DNA, equal parts pCDNA-2xStrep-3xFLAG-TERT and either pBS-U1-hTR or pBS-U1-TSQ containing plasmids. Following the addition of PEI and transfection plasmids, cells were maintained in culture for three days. Cells were then trypsinized and collected, and approximately one million cells were set aside for downstream interphase DNA FISH. High-molecular-weight DNA was collected from the remaining cells for each condition according to the NEB Monarch High-Molecular Weight DNA Extraction Kit for Cells and Blood (NEB #T3050L) protocol. Extracted DNA was quantified using the Qubit BR dsDNA assay (Invitrogen #Q32853) and a Qubit 4 fluorometer.

## Cell line sourcing

Frozen pellets of human embryonic stem cells for *TERT* knockout and *TIN2* mutant hESCs were generously donated by Dirk Hockemeyer's laboratory and immediately processed for sequencing without restarting culture.

HEK293T cells were sourced from ATCC (ATCC CRL-3216). Parental wild-type subclones for hESCs used to generate PARN knock-out subclones were originally sourced from WiCell (WA01). HG002 cells were sourced from the Coriell Institute (GM27730).

## Reporting summary

Further information on research design is available in the Nature Portfolio Reporting Summary linked to this article.

## Data availability

Sequencing data from cell lines and hESCs has been deposited to NCBI's Sequence Read Archive (SRA) and is available through the accession number PRJNA1039890. For healthy donor and TBD patients, minimal pre-processed binary alignment files containing reads mapping to chromosomal termini are available at the accession number above. Source data are provided with this paper.

## Code availability

Our bioinformatic pipeline for telomere measurement from nanopore long reads is available on github at https://github.com/santiago-es/Telometer[43].

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

## Acknowledgements

We are grateful to the participants in the NCI's Inherited Bone Marrow Failure Syndrome study, without whom this work would not be possible. We thank Dr. Neelam Giri, NCI, for clinical support. We also thank the Stanford Blood Center and the several blood donors whose donations contributed to this work under IRB #13942. We thank ONTs for providing development access to their custom telomere basecalling model for R10 flow cell chemistry sequencing data. We thank the entire Artandi laboratory for their discussions and support. We also thank Tobias Schmidt and Jan Karlseder (Salk Institute) for coordinating manuscripts and sharing their insight. This work is supported by the National Institutes of Health (R35 CA197563, P.I.: SEA; 5R01HL131744-06, PI: D.H.).

## Author contributions

S. Sanchez and SEA conceived the project and methodology. S. Sanchez designed the experiments, wrote Telometer, and analyzed the data. Y.W. generated the *PARN* KO hES cell line, Y.G. cultured *PARN* WT and KO hES cells, and performed the TRF Southern blot of *PARN* WT and KO hES clones. A.M. cultured, prepared, and provided *TIN2* mutant and *TERT* knock-out hESCs in the laboratory of D.H. S. Savage provided peripheral blood DNA from *RTEL1* variant cohort and had previously consented and participated in clinical care for these patients. S. Savage performed flow-FISH and TeSLA on *RTEL1* variant cohort DNA and provided access to subsequent data analysis. W.S. participated in the clinical care of TB32, consented the patient for peripheral blood and bone marrow sample collection, and provided the flow-FISH and exon sequencing results for TB32. A.G. trained and tested the binary

classification model. S. Sanchez and S.E.A. wrote the original draft of the manuscript, and all authors participated in subsequent review and editing.

## Competing interests

S. Sanchez and S.E.A. are listed as inventors on a provisional patent application related to clinical applications of telomere measurement related to this work (STFD-006-P). The remaining authors declare no competing interests.
