## [Peer Review File · Nature Communications]

Digital telomere measurement by long-read sequencing distinguishes healthy aging from diseaseEditorial Note: Parts of this Peer Review File have been redacted as indicated to remove third-party material where no permission to publish could be obtained.

REVIEWER COMMENTS

Reviewer #1 (Remarks to the Author):

In this manuscript, Sanchez SE et al illustrate the capability of digital telomere measurement using Oxford Nanopore long-read sequencing platform. Based on cell lines data, the authors showed that the long-read sequencing method can capture disease related telomere shortening events. Upon overexpression of telomerase, they also detected telomere elongation events using this new method. In the analysis of healthy human donors, age related telomere shortening was observed. And because of differential decrease in first and third quartile, the author suggested that the shortening rate in longer telomeres is faster than shorter ones. Furthermore, the authors indicated that the digital telomere measurement can be used to classify not only symptomatic, but also asymptomatic carrier of TBD patients.

The results are interesting and innovative. However, the current manuscript is missing key details, and the results are not consistent throughout to support the authors' claims. Additionally, the text content is in immediate need of restructuring to improve the comprehensiveness for future readers.

Major comments:

1. The authors indicated that "Telomere-containing reads were identified by aligning telomeric repeats to the chromosomal termini of the recently completed telomere-to-telomere human genome." Given the extreme heterogeneity of subtelomeric region, use of single haploid genome may not be sufficient. More data should be provided to demonstrate its applicability to resolve the subtelomeric regions.

2. Detail of the telomere capture procedure should be described.

3. The TRF data for HEK 293T should be included in Fig. 1 in order to compare the telomere length estimation between TRF and digital mean telomere length estimation.

4. Given the heterogenous length of telomeres, it is surprising that the SE of measurement reach a maximal precision of 30-40 base pairs using just 1000 telomere containing reads (Fig. 1h). How many independent replicates in different nanopore flow cell have been sequenced? For the Bootstrapping analysis, how many total reads were used for simulation? Is the Bootstrap resampling analysis performed with replacement or without replacement?

5. The human ESCs are known to have telomere length round 10-14kb {Thomson, 1998 #2108;Rivera, 2017 #2109}. However, in Fig. 2a, the ESCs with WT TIN2 only have a mean telomere length around 5kb? Similar results are shown for Fig. 1j. Can the author confirm the results are generated from ESCs with WT TIN2? Please use the ES cells such as WA01 and WA09 as control. If this is the case, can the author explain such discrepancy.

6. For Fig. 2c, increased short telomeres correlated with days post TERT knockout is understandable and expected. However, why there are more short telomeres (0-1000) in WT than TIN2 284R/+ and TIN2 284R/284R; likewise, why TIN2 284R/+ cells have more extreme short telomeres (0-1000) than TIN2 284R/284R.

This is very confusing as the data also show that the distribution of shorter telomeres (1000-2000) in TIN2 284R/284R>TIN2 284R/+>WT.

Similar results were shown in Fig. 3h, younger individual seem to have more telomeres that are (0-1000) in length, while less telomeres that are (1000-2000) and so on. These results seem to contradict each other. This is unexpected and please elaborate more on this.

7. For Fig. 2i, can the author indicate whether the graph shows the p arm or q arm for each chromosome? Why not both chromosomal ends? It is known the telomere length of two alleles from the same chromosomal end can differ dramatically. Can the author provide the results in allele specific manner to support their conclusion?

8. The authors indicated that "Figure 2 results indicated that chromosomes with shorter telomeres in the control group (GFP) showed a greater magnitude of elongation relative to chromosomes with longer telomeres when treated with telomerase overexpression (Fig. 2h)."

This conclusion maybe true when you only compare Chr. 3 to Chr. 10. However, one can obviously see that: while Chr. 7 has longer telomere length than Chr. 20 in cells transfected with GFP; the telomere elongation for Chr. 7 is obviously more dramatic than Chr. 20 in cells transfected with hTERT and hTR/TSQ1. In addition, there seem to be a lot of discrepancy in the telomere elongation upon hTERT and hTR/TSQ1 overexpression, as the mean telomere length for Chr. 16, 17, and 19 in cells transfected with hTERT and hTR/TSQ1 seem to be shorter than those in cells transfected with GFP control. These discrepancies could be cause by the lack of sequencing depth? Can the author indicate the read number for each chromosome on the graph?

9. The authors indicated that "Quantification of the fraction of telomeres of various sizes in each aging cohort shows that the fraction of the distribution comprised of shorter telomeres increases with age as the longer telomere fraction shrinks with the notable exception of the two shortest fractions which appear to be relatively stable with age in PBLs (Fig. 3h)."

Can the authors indicated which two shortest fractions which appear to be relatively stable with age in PBLs?

10. The authors indicated that "this method is high throughput."

How many samples have been pooled into one flow cell for sequencing? Please also explain the methodology in detail.

11. The authors indicated that "Telomere measurements made by long-read sequencing are comparatively much richer than TRF, flow-FISH while requiring less input DNA than either existing technique and are not limited by polymerase processivity like PCR-based methods STELA and TeSLA."

How much input DNA is used for Oxford Nanopore sequencing in this manuscript? What is the expected output of telomeric reads with that input?

12. For Fig. 3g, please provide detailed analysis steps including the underlying raw data matrix used to generate the PCA plot, and please provide detailed explanation for this figure in the main text. It is obvious that the first two PCA components are sufficient to differentiate the 3 subgroups. Therefore, it seems rather redundant for the need to train a machine learning model to do classification for such small sample size (as illustrated in Fig. S8). To demonstrate the usefulness of the machine learning model, perhaps a larger sample size such be utilized to project its application.

13. On page 5, the authors described that "In RTEL1 patients and carriers, we no longer observe a difference in the extent of decreasing telomere length in the first and third telomere length quartiles with age (Fig. 3b)"

However, Fig. 3f shows that there is a decreasing trend of first and third telomere length quartiles with age. Can authors please explain the discrepancy shown between Fig. 3b and 3f?

14. In Fig. 3b, DC1 and DC2 are two samples collected from same individual with 5 years apart. While it is found that the fraction of smallest telomere (0-1kb) is dramatically increased from 10% in DC2 to

20% in DC1, the median telomere length of DC1 (age 50) seems to be longer than DC2 (age 45). This seems to contradict the ability of digital telomere measurement to detect telomere shortening with age as indicated by Fig. 3a.

15. In Fig. S3 (bottom plot), the spread of subtelomere length seems too big. Is that because of low sequencing depth? If so, please refine the length estimates by including more reads following the suggested depth in the simulation study (Fig. 1h).

Minor comments:

1. For Figure 2, the text indicated that "(c) Stacked bar graphs of telomere length fractions from hESCs with TIN2 mutations by genotype or (d) days post TERT-knockout."

Please correct the text, as (c) is showing days post TERT-knockout, and (d) is showing mutations by genotype.

2. Transient transfection with either hTR or TSQ1 and TERT substantially increased in vitro telomerase activity and resulted in a 1000 bp increase in the mean telomere length after 3 days in culture (Fig. 2, e to h).

It should be (Fig. 2, f to i).

3. The authors indicated that "Figure 2 results indicated that "chromosomes with shorter telomeres in the control group (GFP) showed a greater magnitude of elongation relative to chromosomes with longer telomeres when treated with telomerase overexpression (Fig. 2h)."

It should be Figure 2i instead of 2h.

4. The authors indicated "We confirmed telomeres from these individuals were significantly shorter relative to healthy 35 individuals in their age group (Fig. 3B to C, 3F) and observed a significant correlation ($R^2=0.84$) with mean telomere length measurements made by flow-FISH, the clinical standard method for telomere measurement in peripheral blood (Fig. 1h).

Do they mean results in Fig 1f?

5. The authors indicated that "In RTEL1 patients and carriers, we no longer observe a difference in the extent of decreasing telomere length in the first and third telomere length quartiles with age (Fig. 3b)."

Do they mean Fig. 3i?

6. The results shown in Fig. S4, S5 and S6 are not mentioned in the text.

7. The authors indicated that "Surprisingly, in every sequencing experiment we have performed in either humans or cultured cell lines, including embryonic stem cells, we have measured telomeres as short as dozens of base pairs."

Such extreme short telomeres may be due to DNA fragmentation during genomic DNA purification.

8. The authors concluded that "Furthermore, these data support a model of age-associated telomere attrition where limiting telomerase and not negative selection against cells with very short telomeres is the primary driver behind the accumulation of shorter telomeres with age."

Both negative selection and limiting telomerase are likely to drive this phenotype as the author also quoted that "After an extended period of telomere shortening, a subset of the shortest telomeres become dysfunctional or uncapped, signaling a DNA damage response that triggers replicative senescence, autophagy, or apoptosis (7, 8)."

9. The authors indicated that "TRF was found to systematically overestimate mean telomere length by one to three thousand basepairs in all samples for which both digital and analog telomere measurement were performed."

Different enzyme combinations are known to trim the subtelomeric regions differently. What enzymes are they refer to that "overestimate mean telomere length by one to three thousand basepairs"?

Reviewer #2 (Remarks to the Author):

Summary

=====

In this study, Sanchez et al presents an interesting study where the authors developed an approach to enrich telomeric sequences for long-read sequencing. Using this approach which they developed, they measured changes in telomere length in samples with genetic mutations/alterations in genes involved in telomere maintenance, and in healthy individuals of different ages. They also applied their bioinformatics pipeline to study telomere length in a previously published long-read sequencing dataset of colorectal cancer patients. Further, they had developed a machine learning model to distinguish healthy versus diseased individuals.

A key strength and interest in the work in the application of the method that the authors developed to samples with defects in different components of the telomere maintenance pathway (TIN2, PARN, TERT, RTEL1). However, based on how the data was presented, it isn't entirely clear to me that there is indeed a difference in telomere length distribution between healthy donors and diseased individuals (i.e. individuals with RTEL1 mutations). Additionally, I have very significant concerns about the machine learning model presented given the small cohort size. It is thus not clear how generalizable the machine learning model that the authors had trained to distinguish healthy individuals from those with telomere biology disorders is.

Major points

=====

- 1) In this study, the authors made the claim that telomere length distribution differs between healthy individuals and disease (Figure 3). However, it is really difficult to observe a difference in telomere length between these two groups of individuals from the authors data in Figure 3a and 3b. Is the difference in the distribution of telomere length between both groups statistically significant? What is the magnitude of this difference?
- 2) I think a major interest in the author's work is the application of the method that they had developed to samples with defects in different components of the telomere maintenance pathway. However, it is rather difficult to cross compare samples with defects in each of these components from how the data was presented. The authors may want to reorganize some of their data/plots to make this point clearer.
- 3) Figure 3e and 3f – It might be clearer to plot the healthy and diseased groups into a single scatter plot with the points labelled with different colors. This would make it easier for the readers to assess if there is a difference between both groups.
- 4) Figure 2i – Why were p- and q-arms labels not included here? Also, how reliable is the approach for assigning telomeric reads to each of the chromosomal arms? For instance, can the authors present information on the number of reads observed for each chromosomal arm, which can give an indication as to whether the telomeric reads are misassigned to the wrong chromosomal arm?
- 5) The authors adopted a restriction digestion-based approach to enrich telomeric sequences. I

wonder if the same restriction site was used across all telomeric reads from the same chromosomal arm (i.e. the telomeric reads all end at the same restriction site)? Was this restriction site the closest possible restriction site to the telomeres? If not, does this indicate that digestion of DNA in the author's protocol is incomplete, or that these sites are blocked by DNA modifications? If there is heterogeneity in the restriction site used for the same chromosomal arm, is there a difference in telomere length measurements when different restriction cut sites are used?

6) The authors had defined the telomere-subtelomere boundary as the "final two telomeric repeats before encountering non-telomeric motifs" based on reference 26. Was this done for each read separately? If a longer subtelomeric sequence was captured from the long read, would that then cause a different telomere-subtelomere boundary to be defined for the same chromosomal arm?

7) I think the readers would be quite interested in the performance of the telomere sequencing method, which the authors developed, versus other approaches to enrich and sequence telomeres with long-reads. With this, I think it would be quite helpful if the authors can calculate and present some metrics from the sequencing data that they had already generated to help the readers better assess the performance of each of these methods. Some information which I think will be helpful to provide are:

a) Total number of reads in each sequencing run, number of telomeric reads, number of non-telomeric reads, etc.

b) For the telomeric reads, it would be helpful to provide a breakdown of read from the G- vs. C-strand, how many of these reads were fully telomeric, how many of these reads has the barcoded telomerette on one end of the read and/or subtelomeric sequences on the other end, etc.

c) It may also be helpful to provide a brief description of the non-telomeric reads in these sequencing libraries (e.g. which part of the genome did they come from?).

8) Page 6, line 7 - "including more rapid shortening of the third quartile relative to the first". The p-value of the third quartile ($p=0.14$) is non-statistically significantly. I would refrain from making this statement.

9) Fig 3g - It is not clear to me what raw data was used to generate this PCA plot. To perform a PCA analysis, one will typically need multiple measurements of each sample (e.g. expression of multiple genes of each individual). However, it is not clear to me what measurements of each sample was used for the PCA analysis. If the mean telomere length of each sample alone was used, it would not have been possible to perform a PCA analysis.

10) The authors had built a neural network model to classify samples into diseased vs. healthy individuals. The concern with a neural network is that it often requires lots of data to train, and can potentially overfit for small datasets? I therefore wonder if a simpler logistic regression model might work better in this context.

11) The authors had also mentioned that a "randomized validation set" was used during training (page 15, line 7). Was this resampled from the training/test data? In which case, does this mean that the validation data overlaps with the training data?

12) Figure 3j - It would appear to me that only 14 healthy individuals and 7 individuals with RTEL1 mutations (3 diseased and 4 carriers) were used to develop the machine learning model? This sample size therefore seems too small, and it is therefore not clear how generalizable this model is. When comparing healthy to diseased individuals to carriers, was the comparison done only between 14 healthy vs. 3 carriers? Also, it is not clear to me how the ROC curve was generated from the text. To generate an ROC curve, it is necessary to vary the cutoff value of a "score". However, it is unclear to me what cutoff value was varied and how it was derived. Overall, I think the results on the machine learning model (which is not as critical for the whole story) in this manuscript needs much further support and substantiation.

13) Page 6 Lines 36-40 - The authors stated that telomerase preferentially elongates telomeres at the shortest telomeres based on their observation that chromosomes with shorter first quartile telomere length elongates more. A stronger way to demonstrate this could be to analyze the length of canonical telomeric repeats (TTAGGG - representing telomeric repeats that were initially there) and length of variant telomeric repeats (TTGCGG - representing newly added telomeric repeats) in each telomeric long-read following hTERT and TSQ1 expression as they authors had done in Figure 2f-h.

14) Page 6, Line 31-36 - The authors made the claim that TRF and Flow-FISH overestimates telomere

length. The alternative interpretation is that the experimental/analytical approach that the author developed underestimates telomere length? How do you distinguish between both cases?

15) The font for the axes and labels for some of the figure panels is too small and not very readable (e.g. 2b, 2e, 2h, 2i, 3d, 3e, 3f, 3g, etc.). This makes it very difficult to follow the work as a reader.

Minor points

=====

- 1) Figure 1b – The result showing similar telomere length distribution in telomere capture and WGS is quite interesting.
- 2) Figure 2b – The control is missing. What is the telomere length distribution at Day 0 of TERT KO, or in cells without TERT KO?
- 3) Figure 3a,b – It is really hard to read the age based on the color. It might be helpful to just indicate the actual age below the boxplot.
- 4) Figure 3b. It is difficult to see if an individual belongs to the carrier (solid line) or diseased group (dotted line). The authors may want to highlight these two groups differently (e.g. different colors?)
- 5) Figure 3e,f – The equations are missing the y-intercept values.
- 6) Figure 4a – It is hard to tell which pair of samples display an increase or decrease in telomere length. The authors may want to group samples showing an increase, and those with a decrease in telomere length together.
- 7) Figure 4b and 4c. These results don't look particularly strong and are mostly non-statistically significant. I would recommend removing them or putting these results in the supplement.
- 8) Figure S3 – It seems that the authors have ignored labels of the p- and q-arms of each chromosome when generating these plots. Did the authors aggregate results from the p- and q-arms of the same chromosome in these plots? Why were results from the p- and q-arms aggregated?
- 9) Figure S4f – Are the authors (i) comparing PacBio vs. ONT, (ii) EcoRV vs. HinfI/RsaI/EcoRV, or (iii) their approach with that from citation 26. It is quite confusing given how the plots were labelled.
- 10) Page 12 – Line 9 – “Figure SX”

Reviewer #3 (Remarks to the Author):

In this manuscript, “Digital telomere measurement by long-read sequencing distinguishes healthy aging from disease” Sanchez et al. use digital telomere measurement by nanopore sequencing to better understand how distributions of human telomere length change with age and disease in human cells. They find that human aging is accompanied by a progressive loss of long telomeres and an accumulation of shorter telomeres. Of note, this is not new, but they confirm it with a new method, using machine learning to train a binary classification model that distinguishes healthy individuals from those with telomere biology disorders (this is new). They use data from the Artandi group, who coordinated with Karlseder group. These two groups have a long history of collaboration and top notch work in the telomere field, and have submitted a pair of manuscripts about these ideas.

Together, the two manuscripts now under consideration make valuable contributions to the field. Both are well-written and use the latest in molecular and computational techniques. This is an excellent study with solid information and data, although somewhat of a short report with limited references.

In the discussion they state that telomere measurements made by long-read sequencing are comparatively much richer than TRF and flow-FISH. This is true in that much more information is always gained from cell-by-cell analyses. In this regard, they might also mention (and reference) Telo-FISH, which can also generate TL distributions to monitor short and long telomere populations, and provides more information than TRF on telomere structure and function.

So, while the paper has merit, some additional questions remain that would benefit the paper:

- 1) They developed a sequencing and bioinformatic pipeline (Telometer) capable of reproducibly measuring TL from either whole-genome or telomere-enriched long reads, which will advance our understanding of telomere maintenance mechanisms and the use of TL as a clinical biomarker of aging and disease.
- 2) In the intro, they definitely need to add a reference the recent Telo-sequencing paper. "Recently, it has been demonstrated that long-read sequencing technologies such as PacBio HiFi and Oxford Nanopore (ONT) sequencing can be used to sequence and measure telomeres at unprecedented resolution (26,27)." They missed the first application of ONT data to telomere measurements by the T2T Consortium and the GIAB data (<https://pubmed.ncbi.nlm.nih.gov/34162698/>), as well as the measurement from other human samples (<https://pubmed.ncbi.nlm.nih.gov/33242411/>).
- 3) Digital mean telomere length measured by long-read sequencing was highly correlated to existing gold-standards, TRF Southern blot and flow-FISH, but this should also be validated with an orthogonal sequencing method to get base-level accuracy.
- 4) Did the authors observe any variation in the length of the sub-telomeric regions, which are also known to harbor variation?
- 5) Did they see any non-canonical telomere repeat variants in their data? This has been reported and would be expected here as well.
- 6) They used a training-to-test ratio of 7-3 for their binary machine learning classifier, but this will need to be validated on a larger, and independent cohort. Also, any differences and confounding variables should be added to the model and tested, such as the differences by sex, medical status, drugs, or background.
- 7) The code and methods for the classifier are not available for review, and this is required for new methods.

We have now addressed all the reviewers' queries and I would like to thank each reviewer for their thoughtful and helpful comments. To summarize the new data we have generated for this resubmission, we have done the following:

- Generated deeper sequencing of HEK293T + GFP (Control) cells to increase chromosome-specific coverage across the genome
- Sequenced telomeres from HG002 cells sourced from the Coriell Institute in parallel to Karlseder lab companion manuscript and shown that the telomere lengths match those obtained independently from the Karlseder group
- The neural network binary classification model has been replaced by a simplified linear regression binary classification model
- Included additional analyses pertaining to the bioinformatic pipeline in the revision response (these are not currently planned to be included in the main or supplementary text).
- Answered all reviewer queries below

With these changes, the manuscript is significantly improved and I hope the reviewers will now find it acceptable for publication in Nature Communications.

Reviewer #1 (Remarks to the Author):

In this manuscript, Sanchez SE et al illustrate the capability of digital telomere measurement using Oxford Nanopore long-read sequencing platform. Based on cell lines data, the authors showed that the long-read sequencing method can capture disease related telomere shortening events. Upon overexpression of telomerase, they also detected telomere elongation events using this new method. In the analysis of healthy human donors, age related telomere shortening was observed. And because of differential decrease in first and third quartile, the author suggested that the shortening rate in longer telomeres is faster than shorter ones. Furthermore, the authors indicated that the digital telomere measurement can be used to classify not only symptomatic, but also asymptomatic carrier of TBD patients.

The results are interesting and innovative. However, the current manuscript is missing key details, and the results are not consistent throughout to support the authors' claims..

Major comments:

1. The authors indicated that "Telomere-containing reads were identified by aligning telomeric repeats to the chromosomal termini of the recently completed telomere-to-telomere human genome." Given the extreme heterogeneity of subtelomeric region, use of single haploid genome may not be sufficient. More data should be provided to demonstrate its applicability to resolve the subtelomeric regions.

Response: We agree that there is significant heterogeneity in the subtelomeric region; however, for the purposes of telomere alignment and measurement, the haploid hs1/CHM13v2.0 reference genome is sufficient. We have included additional analysis below to demonstrate this. Below, we compare telomeres measured by our pipeline (Telometer) with Oxford Nanopore's proprietary pipeline (TeloSeq) (which we have been able to test via nextflow) using FASTQ reads generated from a HEK293T SuperTERT, HEK293T GFP or HG002 telomere enriched library. ONT's pipeline uses a custom diploid

reference genome based on the HG002 cell line (from the GIAB project) as opposed to the custom CHM13v2.0/hs1 + Stong 2014 subtelomere assembly reference genome used by all other Telometer experiments. To compare the two assemblies, we applied Telometer to reads aligned to the custom HG002 reference genome ONT uses in TeloSeq (it is publicly available on the TeloSeq github page <https://github.com/rob234king/Teloseq>). We test all combinations of TeloSeq or Telometer using both reference genomes and demonstrate that there is very strong agreement between the telomere length distributions generated by all combinations of pipelines and reference genomes across all cell lines. TeloSeq for HEK293T GFP is not shown as these data were generated using an R9 flow cell and TeloSeq has only been optimized to work for R10 telomere base calls.

Pipeline	First Quartile	Median	Mean	Third Quartile	Reference	Cell Line	Total Telomere
Telometer	3,027.25	4,074.00	4,270.65	5,256.75	CHM13v2.0 + Stong 2014 (Haploid)	HEK293T SuperTERT	21,402.00
TeloSeq	3,147.00	4,068.00	4,220.48	5,138.00	HG002 (Diploid)	HEK293T SuperTERT	2,641.00
Telometer	3,073.00	4,141.00	4,238.21	5,317.00	HG002 (Diploid)	HEK293T SuperTERT	13,105.00
Telometer	4,125.00	5,039.00	5,115.26	6,169.75	HG002 (Diploid)	HG002	7,040.00
Telometer	4,209.00	5,064.00	5,307.37	6,110.00	CHM13v2.0 + Stong 2014 (Haploid)	HG002	11,825.00
TeloSeq	4,292.00	5,123.00	5,294.49	6,049.00	HG002 (Diploid)	HG002	1,837.00

The reference genome and pipeline used can affect the number of telomeres measured but does not affect the telomere measurement distribution (presumably as long as there are sufficient telomere measurements, and in this case all examples are well above the point of diminishing returns for the standard error of the mean, as depicted in Figure 1). In general, it appears that using a more inclusive reference genome (like CHM13v2.0 + Stong 2014) leads to more telomere measurements, despite being haploid.

2. Detail of the telomere capture procedure should be described.

Response: The protocol for telomere capture library preparation used to generate the data in this paper is detailed in the Methods section, copied below. Additionally, the github page for Telometer (<https://github.com/santiago-es/Telometer>) includes an annotated, bench-friendly version of the protocol which will be updated periodically as the protocol evolves and improves. From our Methods section:

Telomere capture sequencing nanopore library preparation

Barcoded telomere capture oligos were annealed to sequencing tether (seqTether) by mixing equimolar amounts of both oligos in low TE buffer, heating to 95°C for 2 minutes and then being allowed to cool at room temperature for an hour. Approximately 3 ug of HMW genomic DNA was ligated to barcoded, freshly duplexed oligos in a 100 µL ligation reaction (10 µL 10X rCutSmart buffer, 5 µL 5 µM duplex capture oligos, 2 µL 2000U/µL T4 DNA ligase, 1 µL 10 mM ATP, 3 µg gDNA, nuclease-free H₂O up to 100 µL) overnight at 37°C. The following day the ligation reaction was heat inactivated at 65°C for 10 minutes. Potential gaps between the capture oligo and the double-strand/single-strand junction were then filled in using the same reaction tube by adding 2 µL (4U) *Sulfolobus* DNA Polymerase IV (NEB #M0327S), 12 µL 10X ThermoPol Buffer, 1 µL 20 mM dNTPs, 1 µL 10 mM ATP, and 4 µL of nuclease-free H₂O followed by incubation at 56°C for 2 minutes and then 72°C for 15 minutes with shaking (500 RPM). Promega ProNex size selection beads were then added to the ligation reaction at a bead-to-solution ratio of 1.6 and the solution was equilibrated on a rotating mixer for 5 minutes at room temperature. The bead solution was then pelleted on a magnet, washed twice with 80% ethanol, allowed to dry on the magnet for 3 minutes, and then eluted in 60 µL of ONT elution buffer for 15 minutes at 37°C. The complete volume of eluted capture-oligo ligated DNA was then ligated onto the ONT sequencing adapter (AMII for R9, NA for R10 libraries) per ONT's barcode ligation sequencing protocol (5 µL sequencing adapter mix, 20 µL 5X Quick T4 DNA Ligase Buffer, 10 µL Quick T4 DNA Ligase, 5 µL nuclease-free H₂O) at room temperature for one hour. The rest of the library preparation and sequencing protocol is performed as for whole-genome sequencing above.

3. *The TRF data for HEK 293T should be included in Fig. 1 in order to compare the telomere length estimation between TRF and digital mean telomere length estimation.*

Response: The manuscript already describes direct comparisons using the same genomic DNA preparation in our 14 healthy individuals and the two samples from the peripheral blood and bone marrow preparations for patient TB32 (TB32-B and TB32-M, respectively) in figure S1. These DNA were treated side-by-side using the same amount of input DNA (10 µg) with the same restriction enzyme cocktail (Hinfl / RsaI) for the same duration of genomic digestion (~16h) since TRF results can vary significantly due to these factors. It is this TRF that the comparison figure (Fig. 1d-e) is based on.

We have since performed TRF on the HEK293T genomic DNA in Figure 1 as described above and include the figure alongside this response. In comparing this TRF with the digital telomere measurements already reported in the manuscript, we similarly observe that TRF overestimates the mean telomere length by approximately 2 kb, in agreement with our other results.

4. Given the heterogenous length of telomeres, it is surprising that the SE of measurement reach a maximal precision of 30-40 base pairs using just 1000 telomere containing reads (Fig. 1h). How many independent replicates in different nanopore flow cell have been sequenced? For the Bootstrapping analysis, how many total reads were used for simulation? Is the Bootstrap resampling analysis performed with replacement or without replacement?

Response: We have found that our method is highly reproducible across separate sequencing runs. To give one example, the top left panel of the figure below shows telomere length distributions from HEK293T cells transiently transfected with GFP from the manuscript’s original sequencing run and from a new sequencing library prepared from the same genomic DNA extract as the original sequencing run produced for this revision of our manuscript. The summary statistics (top right panel) show strong agreement between both sets of measurements.

The bootstrapping analysis in the manuscript (Fig. 1H) was performed with replacement using a dataset containing ~1300 telomeres. The result described is reproducible using any suitably large dataset with the same bootstrapping conditions. To demonstrate this, we have rerun the bootstrapping analysis using a set of over 21,000 telomeres from an HG002 telomere capture sequencing experiment and similarly observe an inflection point for the standard error of the mean of the telomere length distribution near 250 telomere measurements and a plateau near 1000 telomeres which decreases very slowly with additional telomere measurements. The bottom right panel shows the entire curve produced by the bootstrapping analysis while the bottom left zooms in to facilitate comparison to Fig. 1H.

Finally, we collaborated with the Karlseder lab to sequence independent samples of HG002 cells sourced from the Coriell institute to compare our techniques on the same cell line. In the Karlseder lab's manuscript co-submitted alongside ours, the summary results are below:

Manuscript	25th Percentile	Median	Mean	75th Percentile
Schmidt, et al. (Karlseder, TeloSeq)	4011	4979	5247	6115
Sanchez, et al. (Artandi, Telometer)	4209	5064	5307	6110

We believe this demonstrates the technical reproducibility of telomere measurement by long-read sequencing even when using distinct sets of telomere capture oligos, distinct analysis pipelines, and in orthogonal laboratory conditions with different operators to a much greater degree than any other telomere measurement method to date.

5. The human ESCs are known to have telomere length round 10-14kb. However, in Fig. 2a, the ESCs with WT TIN2 only have a mean telomere length around 5kb? Similar results are shown for Fig. 1j. Can the author confirm the results are generated from ESCs with WT TIN2? Please use the ES cells such as WA01 and WA09 as control. If this is the case, can the author explain such discrepancy.

Response: As the reviewer mentions, individual human embryonic stem cell cultures have an intrinsic telomere length "set point" that is maintained during extended culture as a stable average telomere length. However, there is variation in mean telomere length between ES cell cultures. The cells studied here are derived from WIBR3 hES cells, with subclones maintaining a stable telomere length ~7kb as measured by TRF (following overnight MboI/AluI digestion) for 150 days in culture (Choo et al 2022, Reference 31 in the manuscript). As in Figure 1 panels I and J, we consistently find that TRF measurements of telomere length overestimate mean telomere length by approximately 2-3kb (when digested overnight with HinfI/RsaI), likely due to undigested subtelomeric regions which are retained during restriction digest of the end-terminal chromosomal fragments. Together, this demonstrates the telomere length data for this hES culture is internally consistent.

Below, we have included Sanger genotyping of the DC cluster to demonstrate the targeted mutations at the TIN2 locus in the cells sequenced for this manuscript:

tgggcctccactaggggaggccataaggagcgccccacagtcatgctgtttcccttaggaatctcggc
 acccggagggtgatcccccctcgggtatttctctcgcggggtgtcagtacgacaaagggaatccttagagccg

W A S T R G G H K E R P T V M L F P F R N L G

TERF1-interacting nuclear factor 2 isoform 1 →

tgggcctccactaggggaggccataaggagcgccccacagtcatgctgtttcccttaggaatctcggc
 TGGGCCTCCACTAGGGGAGGCCATAAAGGAGCGCCC**TAGG**GTCATGCTGTTTCCTTTAGGAATCTCGGC

Homozygous

TGGGCCTCCACTAGGGGAGGCCATAAAGGAGCGCCC**NANN**GTCATGCTGTTTCCTTTAGGAATCTCGGC

Heterozygous

TGGGCCTCCACTAGGGGAGGCCATAAAGGAGCGCCCCACAGTCATGCTGTTTCCTTTAGGAATCTCGGC

Wild-Type

Regardless of parental cell line, the purpose of this experiment was to show that long-read sequencing of telomeres robustly detects the altered telomere length set point of an isogenic stem cell line with a mutation in the TIN2 DC cluster. A previous publication has demonstrated that this mutation produces approximately a 1-1.5kb reduction in telomere length which is stable over time (Choo et al 2022). This is recapitulated in our long-read sequencing data.

Similarly, the hESC WT cells shown in Fig. 1J are from the WA01 background and the accompanying TRF (performed with overnight HinfI/RsaI digestion) are presented side by side. Our laboratory has heavily utilized this cell line over many passages for a variety of experiments and it is possible that over that time changes have occurred which altered the telomere length of the wild-type cell line, but it is the appropriate control as the parental line of the hESC PARN KO cells which are presented alongside it in both the digital and analog telomere measurement experiments with the goal of recapitulating the degree of telomere shortening observed in individuals with genetic PARN defects. It is likely that other ES cell populations will have differing telomere lengths but reporting on such variation is outside the purpose of this study.

6. For Fig. 2c, increased short telomeres correlated with days post TERT knockout is understandable and expected. However, why there are more short telomeres (0-1000) in WT than TIN2 284R/+ and TIN2 284R/284R; likewise, why TIN2 284R/+ cells have more extreme short telomeres (0-1000) than

TIN2 284R/284R...This is very confusing as the data also show that the distribution of shorter telomeres (1000-2000) in TIN2 284R/284R>TIN2 284R/+>WT.

Response: Thank you for this observation. We also noticed this phenomenon, but do not yet understand the biological basis for this observation so did not comment explicitly in the text. We are interested in pursuing the nature of this phenotype in the future and believe it could be related to the fact that unlike genotypes which specifically impair the telomerase enzyme, the TIN2 284R mutation operates through a poorly understood mechanism. We included this experiment to illustrate two points: first, that as with PARN KO, we can recapitulate the telomere phenotype of a known DC genotype; second, that we can see progressive shortening in a manner that is dependent on the dosage of the mutant allele which could not be previously appreciated with TRF alone (Choo, et al. 2022, Reference 31). It will be interesting to continue to explore the telomere biology of TIN2 and shelterin using long-read sequencing, but we believe that doing so is outside of the scope of this manuscript introducing the capabilities of the method for the first time.

Similar results were shown in Fig. 3h, younger individual seem to have more telomeres that are (0-1000) in length, while less telomeres that are (1000-2000) and so on. These results seem to contradict each other. This is unexpected and please elaborate more on this.

Response: We do not believe that the difference in the 0-1000 bp fraction between the various healthy donor groups is significant and the difference that is being observed is likely due to the small population studied in this manuscript. The linear regressions of the various telomere length summary statistics versus age in this cross-sectional cohort also demonstrate that the relationship between age and length of shorter telomeres is less steep than it is for longer telomeres. Additionally, previous epidemiological studies of telomere length in much larger cohorts have observed significant variation between healthy individuals, which we also observe in this small cohort. The power of our method we are attempting to highlight here is that statistically significant relationships can still be gleaned from the data despite the small sample size (as illustrated in Fig 3E-F) and that the slope of the correlation between age and telomere length varies for different summary statistics of the telomere length distribution.

7. For Fig. 2i, can the author indicate whether the graph shows the p arm or q arm for each chromosome? Why not both chromosomal ends? It is known the telomere length of two alleles from the same chromosomal end can differ dramatically. Can the author provide the results in allele specific manner to support their conclusion?

Response: This figure originally aggregated both arms. Given that it is true there can be variation at the level of individual chromosome arms, we have addressed this in the response to the next comment.

8. The authors indicated that “Figure 2 results indicated that chromosomes with shorter telomeres in the control group (GFP) showed a greater magnitude of elongation relative to chromosomes with longer telomeres when treated with telomerase overexpression (Fig. 2h).”

This conclusion maybe true when you only compare Chr. 3 to Chr. 10. However, one can obviously see that: while Chr. 7 has longer telomere length than Chr. 20 in cells transfected with GFP; the telomere elongation for Chr. 7 is obviously more dramatic than Chr. 20 in cells transfected with hTERT and hTR/TSQ1. In addition, there seem to be a lot of discrepancy in the telomere elongation upon hTERT and hTR/TSQ1 overexpression, as the mean telomere length for Chr. 16, 17, and 19 in cells

transfected with hTERT and hTR/TSQ1 seem to be shorter than those in cells transfected with GFP control. These discrepancies could be caused by the lack of sequencing depth? Can the author indicate the read number for each chromosome on the graph?

Response: As discussed in response to major comment number 4, the original HEK293T GFP data had only around ~1000 telomere reads, which at the level of specific chromosomes translated to very low coverage in the GFP sample of certain chromosomes, and as discussed the aberrant HEK293T genome means the chromosome and chromosome arm dosage / cell is not as is expected for a normal diploid genome. We did not take this into consideration when originally presenting these data. To correct this, we have re-sequenced a new extraction of HEK293T GFP DNA from live cell suspensions previously aliquoted at the collection timepoint of this experiment which were in long-term liquid nitrogen storage, obtaining over ten-thousand more telomere reads. The raw counts per chromosome and chromosome arm are shown below. As is apparent from the data, the counts per chromosome and chromosome arm are not uniform for this aneuploid genome, but they are in similar proportion for each chromosome arm across both the original SuperTERT data (which already contained tens of thousands of telomeres) and the revised HEK293T GFP data. We have also included a new figure showing the boxplots for each chromosome (this time also separated by arm).

While we still believe there is evidence of a trend for shorter chromosome arms to experience more elongation, we take the reviewer's point that there could be additional complexities involved in illustrating this hypothesis definitively which fall outside the intended scope of this manuscript. We have amended the text in the relevant section to read:

"We observed variability in the amount of telomere elongation experienced by different chromosome arms after transient telomerase overexpression (Fig. 2I)." And edited this discussion to highlight that our data in healthy human aging also supports a model of preferential telomerase activity at the shortest telomeres, as other studies have proposed.

Transfection Plasmids  GFP  hTERT + hTR/TSQ1

9. The authors indicated that “Quantification of the fraction of telomeres of various sizes in each aging cohort shows that the fraction of the distribution comprised of shorter telomeres increases with age as

the longer telomere fraction shrinks with the notable exception of the two shortest fractions which appear to be relatively stable with age in PBLs (Fig. 3h)."

Can the authors indicated which two shortest fractions which appear to be relatively stable with age in PBLs?

Response: The 0-1000 and 1000-2000 bp fractions. The text was edited to clarify this point.

10. The authors indicated that "this method is high throughput."

How many samples have been pooled into one flow cell for sequencing? Please also explain the methodology in detail.

Response: In the experiments contained in this manuscript we have successfully pooled up to 8 samples on a single PromethION flow cell. The methodology for telomere capture is detailed in the Methods section "Telomere capture sequencing nanopore library preparation", and language has been added to explain barcode pooling:

For barcoded, pooled libraries up to 8 samples were pooled prior to sequencing adapter ligation at equal molecular weights up to a total of 1000 ng. Sequencing adapter ligation and subsequent preparation for flow cell loading is as previously described.

11. The authors indicated that "Telomere measurements made by long-read sequencing are comparatively much richer than TRF, flow-FISH while requiring less input DNA than either existing technique and are not limited by polymerase processivity like PCR-based methods STELA and TeSLA." How much input DNA is used for Oxford Nanopore sequencing in this manuscript? What is the expected output of telomeric reads with that input?

Response: At least 3 µg of genomic DNA should be used as input for telomere capture libraries, while WGS libraries can be prepared with as little as 1 µg of genomic DNA. With respect to telomere capture, the degree of telomere enrichment and expected output depends on the input used, as has been described by ONT in their TeloSeq release:

[redacted]

For the telomere capture libraries represented in this manuscript, we routinely and consistently used 5 µg of DNA as input, and the relative yield of telomere measurements per gigabases sequence varied as illustrated in Fig. 1C.

12. For Fig. 3g, please provide detailed analysis steps including the underlying raw data matrix used to generate the PCA plot, and please provide detailed explanation for this figure in the main text. It is obvious that the first two PCA components are sufficient to differentiate the 3 subgroups. Therefore, it seems rather redundant for the need to train a machine learning model to do classification for such small sample size (as illustrated in Fig. S8). To demonstrate the usefulness of the machine learning model, perhaps a larger sample size such be utilized to project its application.

Response: The raw data matrix used as input for the PCA is reproduced below. The summary statistics of the telomere length distribution and age were used as input.

sample	mean	first_quartile	median	third_quartile	Age
DC1	3,088.41	1,163.25	3,012.50	4,389.00	50
DC2	3,040.89	1,554.00	2,618.00	4,199.00	45
DC3	3,661.53	2,537.50	3,597.00	4,751.00	36
DC4	4,563.35	3,054.00	4,490.50	5,729.00	35
DC5	4,381.68	3,383.00	4,530.00	5,328.00	6
DC6	4,489.95	3,539.00	4,099.00	5,417.00	21
DC7	3,751.70	2,931.00	3,352.00	4,675.00	47
TB32-B	2,885.06	2,021.00	2,940.00	3,596.00	60
TB32-M	3,263.86	2,322.50	3,054.00	4,043.50	60
ma01	4,931.01	3,643.00	4,622.00	6,101.00	46
ma04	4,593.73	3,369.00	4,352.00	5,381.00	65
ma06	5,334.80	3,884.00	5,098.00	6,254.00	50

ma13	4,862.85	3,609.00	4,634.00	5,863.00	46
ma20	4,788.37	3,439.50	4,271.00	5,721.00	48
ma66	5,249.87	3,909.00	5,072.00	6,483.50	35
oa51	4,496.32	3,645.50	4,086.00	5,511.00	73
oa60	4,637.12	3,427.00	4,357.00	5,729.50	77
oa68	4,302.34	3,116.50	4,155.00	5,284.00	75
ya40	4,715.79	2,747.75	4,393.00	6,474.00	20
ya41	6,371.66	4,270.50	6,706.00	7,922.00	18
ya45	7,421.13	5,413.00	7,586.00	9,164.25	18
ya55	5,156.13	3,502.00	5,064.00	6,630.75	19
ya56	5,954.76	4,417.00	6,080.00	7,580.75	20

The code (R) used to produce the PCA analysis is reproduced below. `telometer_data` stands in for a telometer output file, `dplyr` is used to produce the summary statistics of the column “telomere_length”, and then `prcomp` is used to perform PCA.

```

—
statistics <- telometer_data %>%
  group_by(sample) %>%
  dplyr::summarise(
    mean = mean(telomere_length, na.rm = TRUE),
    first_quartile = quantile(telomere_length, 0.25, na.rm = TRUE),
    median = median(telomere_length, na.rm = TRUE),
    third_quartile = quantile(telomere_length, 0.75, na.rm = TRUE),
    Age = first(Age)
  )
sample_conditions <- distinct(df, sample, .keep_all = TRUE)[, c("sample", "Phenotype", "Age")]

# PCA with prcomp()
statistics_data <- select(statistics, -sample) # remove sample name so it is not used in PCA
pca_result <- prcomp(statistics_data, center = TRUE, scale. = TRUE)
pca_data <- as.data.frame(pca_result$x[, 1:2])

```

```

names(pca_data) <- c("PC2", "PC1")
# Add back in the sample data, aligning with PCA results
pca_data$sample <- statistics$sample
pca_data <- merge(pca_data, sample_conditions, by = "sample")

# Plot (ggplot2)
p <- ggplot(pca_data, aes(x = PC2, y = PC1, color = Age, shape = Phenotype)) +
  geom_point(size = 3) +
  scale_color_viridis_c(option = "viridis", name = "Age") +
  scale_shape_manual(values = c("Healthy" = 16, "Carrier" = 17, "Diseased" = 18)) +
  geom_text(data = subset(pca_data, Phenotype != "Healthy"),
    aes(x = PC2, y = PC1, label = sample, color = Age), vjust = -0.5, hjust = 0.5, check_overlap =
TRUE) +
  labs(title = "", x = "PC2", y = "PC1") +
  theme_minimal() + theme(text=element_text(size=16))
—

```

We have also amended the text to discuss the PCA directly.

13. *On page 5, the authors described that “In RTEL1 patients and carriers, we no longer observe a difference in the extent of decreasing telomere length in the first and third telomere length quartiles with age (Fig. 3b)”*

However, Fig. 3f shows that there is a decreasing trend of first and third telomere length quartiles with age. Can authors please explain the discrepancy shown between Fig. 3b and 3f?

Response: The original text contains a typo here and the quoted text should refer to Fig. 3E. We also realize the original text as written is unintuitive. Our intention was to describe the differences in the slope of the relationship between cross-sectional Age and the various telomere length summary statistics in healthy individuals or RTEL1 variant patients. We point this out immediately after highlighting that TERT null hESCs experience rapid first quartile telomere length shortening as well as third quartile telomere length shortening while white blood cells from healthy individuals experience a “slower” decrease in the first quartile telomere length relative to the third quartile. We have edited the text to reflect this.

Additionally, in response to a different reviewer’s comment Figures 3E and 3F will be combined and TB32 will now be included in the DC cohort as the patient has now been confirmed to fit the clinical diagnosis of a telomere biology disorder.

14. *In Fig. 3b, DC1 and DC2 are two samples collected from same individual with 5 years apart. While it is found that the fraction of smallest telomere (0-1kb) is dramatically increased from 10% in DC2 to 20% in DC1, the median telomere length of DC1 (age 50) seems to be longer than DC2 (age 45). This seems to contradict the ability of digital telomere measurement to detect telomere shortening with age as indicated by Fig. 3a.*

Response: We disagree that these data are contradictory. It is important to remember that telomere biology disorder patients are not experiencing healthy aging and illustrating the distinction between healthy aging and telomere biology disorders as observed through the quantitative lens of previously inaccessible telomere length distributions is a major contribution of this manuscript. Finally, we believe that the increase in short telomeres over this five-year period is consistent with progression of this individual's disease with age.

That said, we believe that there may be interesting clonal dynamics in this patient's hematopoietic system which could be contributing to this observation, including entering levels of telomere shortening in which negative selection due to telomere shortening mediated senescence is creating a bottleneck for the hematopoietic stem cell pool, and we are continuing to explore these questions in ongoing projects in our laboratory. While the following data will not be included in this manuscript as the relationship between telomere length and clonality is not a focus of this paper, we used cuteSV to characterize structural variants in samples DC2 and DC1 and observed a significant increase in structural variants over a 5-year period. We additionally performed targeted sequencing of a panel of genes commonly involved in the development of clonal hematopoiesis and identified a significant STAT3 mutant clone (VAF ~14%) at age 45 which was no longer present at age 50, suggesting an underlying process of stem cell competition or selection occurring in this patient's marrow. In the figures below, S1 corresponds to DC2 and S2 to DC1. We are interested in continuing to explore these questions with long-read sequencing.

Targeted Sequencing of CH Genes

Sample	Gene	VAF	Mutation Type
S2	TET2	0.99	germline
S2	TP53	0.52	germline
S1	TET2	0.99	germline
S1	TP53	0.53	germline
S1	STAT3	0.14	clone

15. In Fig. S3 (bottom plot), the spread of subtelomere length seems too big. Is that because of low sequencing depth? If so, please refine the length estimates by including more reads following the suggested depth in the simulation study (Fig. 1h).

Response: The purpose of this supplementary figure is to demonstrate that the length of remaining subtelomere on telomeric reads is dependent on the number of restriction enzymes used during genomic digestion, which is itself not particularly surprising or interesting. However, given that TRF and flow-FISH are most likely overestimating telomere length due to undigested subtelomeric DNA that is included on the terminal telomere fragment, we thought it was important to document this quantitatively

using the data we've already generated. The single enzyme EcoRV digestion represented in the bottom of S3 was performed for only 30 minutes - 1 hour, as detailed in the methods section pertaining to telomere capture library preparation, so it is also likely incomplete.

That said, we've refined the estimates by using subtelomere length measurements from an experiment with >21,000 telomeric reads instead of the original ~2,000 reads used in the original S3 EcoRV figure.

Minor comments:

1. For Figure 2, the text indicated that “(c) Stacked bar graphs of telomere length fractions from hESCs with TIN2 mutations by genotype or (d) days post TERT-knockout.”

Please correct the text, as (c) is showing days post TERT-knockout, and (d) is showing mutations by genotype.

Response: We have done so.

2. Transient transfection with either hTR or TSQ1 and TERT substantially increased *in vitro* telomerase activity and resulted in a 1000 bp increase in the mean telomere length after 3 days in culture (Fig. 2, e to h).

It should be (Fig. 2, f to i).

Response: We have corrected these errors.

3. *The authors indicated that “Figure 2 results indicated that “chromosomes with shorter telomeres in the control group (GFP) showed a greater magnitude of elongation relative to chromosomes with longer telomeres when treated with telomerase overexpression (Fig. 2h).”*

It should be Figure 2i instead of 2h.

Response: We have corrected these errors.

4. *The authors indicted “We confirmed telomeres from these individuals were significantly shorter relative to healthy 35 individuals in their age group (Fig. 3B to C, 3F) and observed a significant correlation ($R^2=0.84$) with mean telomere length measurements made by flow-FISH, the clinical standard method for telomere measurement in peripheral blood (Fig. 1h).*

Do they mean results in Fig 1f?

Response: The text has been edited to point to the accurate figures after figure editing.

5. *The authors indicated that “In RTEL1 patients and carriers, we no longer observe a difference in the extent of decreasing telomere length in the first and third telomere length quartiles with age (Fig. 3b).”*

Do they mean Fig. 3i?

Response: We have corrected the text to reflect the updated figures.

6. *The results shown in Fig. S4, S5 and S6 are not mentioned in the text.*

Response: The supplementary figures have been substantially updated and rearranged, but all are now mentioned in the main text or methods section of the manuscript.

7. *The authors indicated that “Surprisingly, in every sequencing experiment we have performed in either humans or cultured cell lines, including embryonic stem cells, we have measured telomeres as short as dozens of base pairs.”*

Such extreme short telomeres may be due to DNA fragmentation during genomic DNA purification.

Response: While fragmentation is certainly possible and likely contributing at some level to extremely short telomeres, we believe the fact that the 0-1000 bp fraction is responsive to genetic perturbation, for example by knocking out telomerase, or directly reducing telomerase activity via PARN KO suggests to us that an appreciable amount of these extremely short telomeres are indeed biological. To make this point more clearly in the text and not focus particularly on telomeres as short as dozens of bp, this sentence has been edited to read:

Surprisingly, in every sequencing experiment we have performed we have measured a number of telomeres shorter than 1000 bp.

8. The authors concluded that “Furthermore, these data support a model of age-associated telomere attrition where limiting telomerase and not negative selection against cells with very short telomeres is the primary driver behind the accumulation of shorter telomeres with age.”

Both negative selection and limiting telomerase are likely to drive this phenotype as the author also quoted that “After an extended period of telomere shortening, a subset of the shortest telomeres become dysfunctional or uncapped, signaling a DNA damage response that triggers replicative senescence, autophagy, or apoptosis (7, 8).”

Response: We do not think that the conclusion highlighted here is contradicted by the statement quoted from the introduction.

There is evidence, such as that cited, that critically short telomeres can signal through the DNA damage response to trigger replicative senescence and the downstream consequences of that process, and there is also evidence that shorter telomeres accumulate with age in humans and with serial passage in BJ fibroblasts with limiting telomerase, as we discuss.

That said, the question of whether negative selection against cells with very short telomeres is happening via replicative senescence in healthy aging remains unanswered. If it is true that the accumulation of short telomeres in healthy aging is due to negative selection, then that negative selection process should still occur in patients with TBDs or in TERT knockout cells. However, in our RTEL1 variant cohort and in the TERT knockout hESCs we observe the shortest telomere fractions increase in similar proportions *without* replicative senescence of the culture or population, as the RTEL1 variant cohort telomeres were measured from leukocytes extracted from the peripheral blood and the TERT knockout hESCs did not reach replicative senescence until days 110-120 post TERT knockout, as we discuss. On the other hand, in our healthy aging cohort we see telomere length distributions shorten over time but the shortest telomere fraction remains relatively constant which can be explained by healthy individuals having sufficient telomerase in the hematopoietic stem cell pool to preferentially maintain short telomeres.

Therefore, unless the TERT knockout hESCs and RTEL1 variant leukocytes developed a *de novo* adaptation to very short telomeres, our data suggest that the accumulation of short telomeres in healthy aging reflects the insufficiency of telomerase in the cell population to indefinitely maintain the “floor” of the telomere length distribution.

9. The authors indicated that “TRF was found to systematically overestimate mean telomere length by one to three thousand basepairs in all samples for which both digital and analog telomere measurement were performed.”

Different enzyme combinations are known to trim the subtelomeric regions differently. What enzymes are they refer to that “overestimate mean telomere length by one to three thousand basepairs”?

Response: For all TRFs shown in this manuscript, genomic digestion was performed overnight with HinfI and RsaI (see: Methods, Telomere Restriction Fragment Southern Blot).

Reviewer #2 (Remarks to the Author):

In this study, Sanchez et al presents an interesting study where the authors developed an approach to enrich telomeric sequences for long-read sequencing. Using this approach which they developed, they measured changes in telomere length in samples with genetic mutations/alterations in genes involved in telomere maintenance, and in healthy individuals of different ages. They also applied their bioinformatics pipeline to study telomere length in a previously published long-read sequencing dataset of colorectal cancer patients. Further, they had developed a machine learning model to distinguish healthy versus diseased individuals.

A key strength and interest in the work in the application of the method that the authors developed to samples with defects in different components of the telomere maintenance pathway (TIN2, PARN, TERT, RTEL1).

Major points

1) *In this study, the authors made the claim that telomere length distribution differs between healthy individuals and disease (Figure 3). However, it is really difficult to observe a difference in telomere length between these two groups of individuals from the authors data in Figure 3a and 3b. Is the difference in the distribution of telomere length between both groups statistically significant? What is the magnitude of this difference?*

Response: We agree that the presentation of this data in Figure 3 could be improved. We have now combined Figures 3A through C into one figure, losing the distinction between symptomatic and asymptomatic RTEL1 variant carriers (but this information is documented in the supplement and we will flag where appropriate). This new figure presents the same data, but this time includes the age in red above the distribution, includes the coded sample names on the X axis, and distinguishes TBD patients from healthy individuals by color. We believe that this presentation of the data makes it clear that TBD patients, as expected, have short telomeres relative to their near-age peers.

In the clinic, telomere biology disorders are diagnosed by obtaining a mean leukocyte telomere length by flow-FISH and comparing telomere length to near-age peers using historically accumulated population-level data on telomere length vs age. That in mind, we do not think it would be meaningful to statistically compare that a 35 year old TBD patient is “significantly different” from a healthy 35 year old, as healthy individuals will also be “significantly different” from each other due to the level of variation in telomere length which is also present in healthy individuals. We therefore opted to use PCA to highlight that the information contained in the entire telomere length distribution in addition to age can meaningfully distinguish known TBD variant carriers from healthy individuals, which is not always possible with flow-FISH.

2) *I think a major interest in the author’s work is the application of the method that they had developed to samples with defects in different components of the telomere maintenance pathway. However, it is rather difficult to cross compare samples with defects in each of these components from how the data was presented. The authors may want to reorganize some of their data/plots to make this point clearer.*

Response: We agree that this is an interesting aspect of the data we present, but it was not our intention to center a detailed analysis of the contributions of various elements of the telomere maintenance pathway in our manuscript, although this is an interesting area of future exploration we would like to pursue using this technology. The data are presented in such a way to first introduce the methodology, demonstrate that it can recapitulate known phenotypes of telomere biology disorders, and then demonstrate its applicability in healthy and diseased humans.

3) *Figure 3e and 3f – It might be clearer to plot the healthy and diseased groups into a single scatter plot with the points labelled with different colors. This would make it easier for the readers to assess if there is a difference between both groups.*

Response: We agree and have done this. Additionally, we now include TB32 as a composite of TB32-B and TB32-M in the TBD patient cohort now that this patient has been clinically confirmed to have a telomere biology disorder. When remaking this figure we also discovered that the healthy patient linear regressions only included 11 of the 14 healthy individuals. This error has been corrected and all 14 healthy individuals in Fig 3A are now included in the linear regression analysis, with updated statistics. The interpretation of the figures remains the same, indeed the difference in the slope of the correlation between 25th percentile telomere length and age between healthy individuals and TBD patients is now even more apparent.

4) *Figure 2i – Why were p- and q-arms labels not included here? Also, how reliable is the approach for assigning telomeric reads to each of the chromosomal arms? For instance, can the authors present information on the number of reads observed for each chromosomal arm, which can give an indication as to whether the telomeric reads are misassigned to the wrong chromosomal arm?*

Response: We have recreated this figure to appropriately disaggregate P and Q arm telomeres for each chromosome. The mapping quality of a read to the reference is given by its MAPQ score, which Telometer reports as a part of the output, but the reliability of mapping all true telomeric reads to the

correct chromosome arm will also depend on the quality of the reference genome and its coverage of the inherent diversity of the subtelomeric region. For this reason, we do not focus on chromosome specific measurements in this manuscript apart from the experiment shown in Figure 2 using transient transfection of HEK293T cells. Because this is a controlled experiment, any mapping mistakes made by minimap2 or defects in the reference genome should affect both the GFP and SuperTERT/TSQ arms of the experiment, aiding in the interpretation of the results. Finally, it is also worth mentioning that HEK293T cells are known to have aneuploid genomes, with large scale structural variation and polyploidy in some chromosomes, which could also affect the number of measurements per chromosome. We have included the counts for each chromosome arm in this experiment in a response to a different reviewer's comment.

5) *The authors adopted a restriction digestion-based approach to enrich telomeric sequences. I wonder if the same restriction site was used across all telomeric reads from the same chromosomal arm (i.e. the telomeric reads all end at the same restriction site)? Was this restriction site the closest possible restriction site to the telomeres? If not, does this indicate that digestion of DNA in the author's protocol is incomplete, or that these sites are blocked by DNA modifications? If there is heterogeneity in the restriction site used for the same chromosomal arm, is there a difference in telomere length measurements when different restriction cut sites are used?*

Response: The purpose of the restriction digest in our protocol is to eliminate some amount of genomic DNA from the library prep, and the reason a blunt cutter is used is to dA tail the non-telomeric blunt end to reduce the likelihood that it is ligated to the sequencing adapter – but the restriction cut site or enzyme used doesn't really matter. Additionally, our protocol only includes restriction digestion with a single cutter for approximately 30 minutes to 1 hour, so the digestion of the genomic DNA could very well be incomplete.

We have repeated sequencing of the HEK293T GFP DNA extract to obtain deeper coverage of chromosome arms for Fig 2i. The original HEK293T GFP experiment was performed with telomere capture without restriction digestion on an R9 cell and the revision sequencing run was performed with EcoRV digestion. This difference reflects the evolution of the library preparation protocol throughout the development of this manuscript. Nevertheless, it is evident from the figure provided in response to Reviewer #1's third comment that this methodological variation, while improving enrichment for telomeric reads, had little to no impact on telomere length measurement.

6) *The authors had defined the telomere-subtelomere boundary as the “final two telomeric repeats before encountering non-telomeric motifs” based on reference 26. Was this done for each read separately? If a longer subtelomeric sequence was captured from the long read, would that then cause a different telomere-subtelomere boundary to be defined for the same chromosomal arm?*

Response: Determination of the telomere-subtelomere boundary is done separately and empirically for every read which aligns to the first or final 15,000 bp of the reference chromosome. If the subtelomeric site at which telomeres begin differs between two telomeres on the same arm, the boundary will be different for both reads, but the length is only determined from the subtelomeric boundary to the end of the telomere. The length of the subtelomeric portion has no impact on the telomere length or subtelomere boundary. To illustrate what this boundary might look like on an individual read, I have included a figure below in which each square corresponds to an individual, telomere-containing read,

the x-axis corresponds to the length of the read from 0 to its arbitrary length, and the y-axis shows the “telomere signal”, defined as the fraction of a sliding 50 bp window which contains the canonical telomere sequence, and finally on top of each square is the telomere length for this read as determined by Telometer. This signal based method is actually how we first attempted to measure telomeres from long-reads, but found its performance was inferior to the method used by Telomap for PacBio (CY Tham et al, 2023) at consistently determining the subtelomeric boundary due to the underlying variation in sub/telomeric sequence, sequencing noise, and the challenge of choosing the optimal parameters (like signal threshold, sliding window size, tolerance for variation, etc.) for a signal-detection based method.

7) *I think the readers would be quite interested in the performance of the telomere sequencing method, which the authors developed, versus other approaches to enrich and sequence telomeres with long-reads. With this, I think it would be quite helpful if the authors can calculate and present some metrics from the sequencing data that they had already generated to help the readers better assess the performance of each of these methods. Some information which I think will be helpful to provide are:*

a) *Total number of reads in each sequencing run, number of telomeric reads, number of non-telomeric reads, etc.*

Response: We have included a supplementary spreadsheet which contains sequencing metadata for the experiments represented in the manuscript for interested readers.

b) *For the telomeric reads, it would be helpful to provide a breakdown of read from the G- vs. C-strand, how many of these reads were fully telomeric, how many of these reads has the barcoded telomerette on one end of the read and/or subtelomeric sequences on the other end, etc.*

Response: There is a bias for the C strand in both WGS and telomere capture, and this bias appears to be more pronounced for telomere capture. Fully telomeric reads would be thrown out by telometer.py and some subtelomeric sequence is required for any read where telomeres are measured. For barcoded multiplex TC experiments, 100% of the measured telomere reads are barcoded. For simplex experiments, the percent of telomeres containing reads in which the full barcode can be identified is ~70% of all measured telomeres in a representative experiment.

c) *It may also be helpful to provide a brief description of the non-telomeric reads in these sequencing libraries (e.g. which part of the genome did they come from?).*

Response: Non-telomeric reads can come from anywhere in the genome and we do not think it would be particularly informative or helpful for the main thrust of our manuscript to detail this. Any fragment of genomic DNA which can be ligated onto the ONT sequencing adapter can be sequenced. Although the library preparation protocol has been optimized for telomere enrichment, both our protocol and ONT's TeloSeq protocol can only produce libraries in which a minority of the reads are "on-target" for the telomere region; nevertheless, the level of enrichment we see is at least several hundred-fold, making it possible to measure a very small fraction (0.02%) of the genome at high coverage.

8) Page 6, line 7 - “including more rapid shortening of the third quartile relative to the first”. The p-value of the third quartile ($p=0.14$) is non-statistically significantly. I would refrain from making this statement.

Response: We have edited the text to reflect a trend that does not reach statistical significance.

9) Fig 3g – It is not clear to me what raw data was used to generate this PCA plot. To perform a PCA analysis, one will typically need multiple measurements of each sample (e.g. expression of multiple genes of each individual). However, it is not clear to me what measurements of each sample was used for the PCA analysis. If the mean telomere length of each sample alone was used, it would not have been possible to perform a PCA analysis.

Response: Please see our response to Reviewer #1 major comment 12 in which we include the raw data matrix used to generate the PCA. Briefly, the summary statistics of the telomere length distribution and age.

10) The authors had built a neural network model to classify samples into diseased vs. healthy individuals. The concern with a neural network is that it often requires lots of data to train, and can potentially overfit for small datasets? I therefore wonder if a simpler logistic regression model might work better in this context.

Response: We have done as the reviewer suggested, please see the dedicated section responding to model comments at the end.

11) The authors had also mentioned that a “randomized validation set” was used during training (page 15, line 7). Was this resampled from the training/test data? In which case, does this mean that the validation data overlaps with the training data?

Response: Yes, the training and test datasets overlapped for the neural network model. Also, please see the dedicated section responding to model comments at the end.

12) Figure 3j – It would appear to me that only 14 healthy individuals and 7 individuals with RTEL1 mutations (3 diseased and 4 carriers) were used to develop the machine learning model? This sample size therefore seems too small, and it is therefore not clear how generalizable this model is. When comparing healthy to diseased individuals to carriers, was the comparison done only between 14 healthy vs. 3 carriers? Also, it is not clear to me how the ROC curve was generated from the text. To generate an ROC curve, it is necessary to vary the cutoff value of a “score”. However, it is unclear to me what cutoff value was varied and how it was derived. Overall, I think the results on the machine learning model (which is not as critical for the whole story) in this manuscript needs much further support and substantiation.

Response: Please see the dedicated section on the binary classification model.

13) Page 6 Lines 36-40 – The authors stated that telomerase preferentially elongates telomeres at the shortest telomeres based on their observation that chromosomes with shorter first quartile telomere length elongates more. A stronger way to demonstrate this could be to analyze the length of canonical telomeric repeats (TTAGGG – representing telomeric repeats that were initially there) and length of

variant telomeric repeats (TTGCGG – representing newly added telomeric repeats) in each telomeric long-read following *hTERT* and *TSQ1* expression as they authors had done in Figure 2f-h.

Response: We agree that this would be a stronger way to illustrate the phenomenon of preferential elongation, and this was our intention when originally conducting this experiment. Unfortunately, we are unable to analyze variant sequences using our method. In data produced from R9 flow cells, telomeric repeats are frequently miscalled in a stereotyped manner, as was detailed in Tan, et al. (Reference 42). In data produced from R10 flow cells, the ONT basecalling models have built-in improved telomere basecalling which increases the fidelity of canonical telomeric repeat sequences but sacrifices the ability to distinguish between canonical and non-canonical telomeres.

We believe the raw sequencing signal from the *TSQ1* variant sequence (TTGCGG / AACGCC) is similar enough to those produced by telomeric repeats to be either miscalled in the same way in R9 flow cells or corrected to canonical telomere sequences by error in R10 flow cells. To explore this, we followed the approach of Tan, et al. (42) by simulating the raw sequencing signal produced by the various telomere motifs on an R9.4 flow cell (the chemistry for the flow cell the *TSQ1* experiment samples were sequenced with) versus the signal produced by the *TSQ1* variant sequence and identified one of the commonly miscalled telomere motifs (CCTTGG) which produces a very similar raw sequencing signal to that predicted for the *TSQ1* variant sequence.

To perform this analysis, we downloaded the R9.4 basecalling model kmer signal data from ONT's public repository (https://github.com/nanoporetech/kmer_models) and simulated the signal produced at each position in a synthetic 120 bp stretch of telomeric motifs containing either the *TSQ1* variant sequence or one of the canonical telomere motifs or their common miscalls. Together, these data suggest that we cannot distinguish the *TSQ1* variant sequences in our data from other telomeric sequences.

That said, the *TSQ1* sequencing experiment was done using telomere capture oligos designed against the variant sequence and as we mention in the text, attempting telomere capture with *TSQ1* variant

capture oligos in cells which did not receive the TSQ1 plasmid did not result in a successful sequencing library (very low pore occupancy was observed and no telomere reads were produced, suggesting that the ONT sequencing adapter was unable to ligate to capture probes, Figure S1). We take this as evidence that the TSQ1 variant sequence was captured by molecules which were then sequenced and produced telomere measurements.

14) Page 6, Line 31-36 - The authors made the claim that TRF and Flow-FISH overestimates telomere length. The alternative interpretation is that the experimental/analytical approach that the author developed underestimates telomere length? How do you distinguish between both cases?

Response: Telomere length underestimation by sequencing is a plausible alternative hypothesis, particularly if the read lengths of the sequencing experiment are very close to or shorter than the expected telomere lengths for a particular sample as can often be the case for PacBio (see figure S4). That said, we believe this explanation to be less likely for ONT long-read sequencing where the sequencing method is much less limited by read length.

One of the motivations behind the TSQ/SuperTERT experiment was to show that ONT sequencing captures complete telomeres by measuring *de novo* elongation. The facts that telomere capture requires the presence of a single-stranded overhang and that we can see *de novo* elongation in a short period approximately of the same magnitude which has been observed using other methods in similar experimental contexts (Cristofari and Lingner, reference 32) both suggest to us we are not significantly underestimating the length of the telomere.

Finally, TRF and Flow-FISH are expected to overestimate telomere length, because both cannot differentiate telomere from subtelomere remaining after restriction digestion. Flow-FISH is calibrated to TRF, and TRF depends on restriction digestion, so both gold-standard methodologies suffer from this limitation.

15) The font for the axes and labels for some of the figure panels is too small and not very readable (e.g. 2b, 2e, 2h, 2i, 3d, 3e, 3f, 3g, etc.). This makes it very difficult to follow the work as a reader.

Response: We have improved the readability of these figures.

Minor points

=====

1) Figure 1b – The result showing similar telomere length distribution in telomere capture and WGS is quite interesting.

2) Figure 2b – The control is missing. What is the telomere length distribution at Day 0 of TERT KO, or in cells without TERT KO?

Response: Unfortunately, sampling Day 0 of TERT KO is not feasible as these data come from a clonally-derived line of TERT-KO cells post Cre-mediated loopout (published in Boyle et al 2020, Figure 2, panel D). As such, by the time a sufficient number of cells have been generated for clonal isolation, genotyping, expansion, and telomere length measurement, telomeres have already been shortening for several weeks. The purpose of our inclusion of this cell line was to measure gradual telomere loss over time in a clonal background with no additional genetic perturbations. While including a measurement of

parental telomere lengths prior to TERT KO would demonstrate the difference in parental telomere length set point relative to the shortened telomeres of the progeny, this is an improper control for this telomere attrition over time due to TERT loss. As Day 66 was the first measured time point for this cell line, it may be considered "Time Point 0" and acts as the control for subsequent timepoints since the only experimental variable is additional time in culture.

3 and 4) *Figure 3a,b – It is really hard to read the age based on the color. It might be helpful to just indicate the actual age below the boxplot. Figure 3b. It is difficult to see if an individual belongs to the carrier (solid line) or diseased group (dotted line). The authors may want to highlight these two groups differently (e.g. different colors?)*

Response: Please see our response to Reviewer #2 major comment #1, we have revamped Figures 3A-C into a single figure with improved readability. The symptom variable is documented in the supplementary table describing the RTEL1 patient cohort.

5) *Figure 3e,f – The equations are missing the y-intercept values.*

Response: These figures have been condensed to a single figure, please see the response to Reviewer #2s 3rd major comment. The y-intercept has also been added.

6) *Figure 4a – It is hard to tell which pair of samples display an increase or decrease in telomere length. The authors may want to group samples showing an increase, and those with a decrease in telomere length together.*

Response: We have edited Figure 4 to improve its readability. We have also edited the text to indicate that the patients highlighted by red boxes represent those for whom the tumor telomere length is greater than or equal to the benign colonic epithelia telomere length.

7) Figure 4b and 4c. These results don't look particularly strong and are mostly non-statistically significant. I would recommend removing them or putting these results in the supplement.

Response: The linear regressions for Figure 4 have been moved to supplementary information but are still described in the text. We believe Figure 4C being statistically insignificant was noteworthy as it is expected that cells whose telomeres are maintained by an oncogenic mechanism like telomerase

reactivation will no longer have a relationship between telomere length and age as other processes now dominate telomere maintenance.

8) *Figure S3 – It seems that the authors have ignored labels of the p- and q-arms of each chromosome when generating these plots. Did the authors aggregate results from the p- and q-arms of the same chromosome in these plots? Why were results from the p- and q-arms aggregated?*

Response: As before, we did aggregate results for both arms and we understand this could potentially mask variation between the arms. That said, for this figure describing the undigested subtelomeric sequence observed in telomeric reads, we do not see the benefit of including chromosome specific information which also suffers from limitations of mapping the subtelomeric region with the reference genome currently at our disposal, as discussed in response to other reviewer comments. S3 has been edited to show the bulk subtelomeric length distribution only.

9) *Figure S4f – Are the authors (i) comparing PacBio vs. ONT, (ii) EcoRV vs. HinfI/RsaI/EcoRV, or (iii) their approach with that from citation 26. It is quite confusing given how the plots were labelled.*

Response: (iii) is the correct interpretation of this figure. Figure S4F is now redundant with Figure S3, and we have removed figure S3 and kept S4F.

10) *Page 12 – Line 9 – “Figure SX”*

Response: This typo has been corrected to refer to “Figure S2”.

Reviewer #3 (Remarks to the Author):

In this manuscript, “Digital telomere measurement by long-read sequencing distinguishes healthy aging from disease” Sanchez et al. use digital telomere measurement by nanopore sequencing to better understand how distributions of human telomere length change with age and disease in human cells. They find that human aging is accompanied by a progressive loss of long telomeres and an accumulation of shorter telomeres. Of note, this is not new, but they confirm it with a new method, using machine learning to train a binary classification model that distinguishes healthy individuals from those with telomere biology disorders (this is new). They use data from the Artandi group, who coordinated with Karlseder group. These two groups have a long history of collaboration and top notch work in the telomere field, and have submitted a pair of manuscripts about these ideas.

Together, the two manuscripts now under consideration make valuable contributions to the field. Both are well-written and use the latest in molecular and computational techniques. This is an excellent study with solid information and data, although somewhat of a short report with limited references.

In the discussion they state that telomere measurements made by long-read sequencing are comparatively much richer than TRF and flow-FISH. This is true in that much more information is always gained from cell-by-cell analyses. In this regard, they might also mention (and reference) Telo-FISH, which can also generate TL distributions to monitor short and long telomere populations, and provides more information than TRF on telomere structure and function.

So, while the paper has merit, some additional questions remain that would benefit the paper:

1) *They developed a sequencing and bioinformatic pipeline (Telometer) capable of reproducibly measuring TL from either whole-genome or telomere-enriched long reads, which will advance our understanding of telomere maintenance mechanisms and the use of TL as a clinical biomarker of aging and disease.*

2) *In the intro, they definitely need to add a reference the recent Telo-sequencing paper. “Recently, it has been demonstrated that long-read sequencing technologies such as PacBio HiFi and Oxford Nanopore (ONT) sequencing can be used to sequence and measure telomeres at unprecedented resolution (26,27).” They missed the first application of ONT data to telomere measurements by the T2T Consortium and the GIAB data (<https://pubmed.ncbi.nlm.nih.gov/34162698/>), as well as the measurement from other human samples (<https://pubmed.ncbi.nlm.nih.gov/33242411/>).*

Response: We have added these citations in the appropriate sections of the text and we thank the reviewer for the suggestion.

3) *Digital mean telomere length measured by long-read sequencing was highly correlated to existing gold-standards, TRF Southern blot and flow-FISH, but this should also be validated with an orthogonal sequencing method to get base-level accuracy.*

Response: We do not believe telomere measurements made by PacBio sequencing to be equivalent to those made by ONT sequencing due to the read length limitations of PacBio sequencing. We demonstrate this using previously published PacBio data (CY Tham et al, Reference 26) in Figure S4. The nature of CCS/HiFi reads is such that reads with sufficient coverage to achieve high levels of base accuracy will be much shorter than the average ONT read, and this is reflected in the relationship between telomere length and read length. Ideally, telomere length measurements should be nearly uncorrelated to read length, but if read length is acting as a bottleneck to telomere length measurement you would expect there to be a strong correlation, which is exactly what is observed in Fig S4 for PacBio, but to a much lower degree in ONT reads. Therefore, for telomere length measurements in cells whose telomeres are mostly between 0 and 7000 bp, PacBio could accurately capture the telomere length distribution, but as our data suggests this is not always the case and there are instances of telomeres close to or greater than 20 kb which could be missed by PacBio.

We have attempted to perform telomere enrichment for PacBio sequencing as described in CY Tham et al but this method did not work well in our hands. The amount of starting material required, time to perform the protocol, and relative cost of PacBio sequencing compared to ONT sequencing lead us to focus on the ONT methodology.

That said, the improved telomere basecalling now built-in to the default dorado R10 high-accuracy basecalling model is able to produce high levels of telomere sequence accuracy. In the figure below we count the frequency of 6 base pair motifs contained in telomere-containing reads from an experiment measuring telomeres in HG002 cells on an R10 flow cell using the latest dorado basecalling model and one can appreciate that the vast majority of all telomeric motifs are made up of the canonical sequences (TTAGGG, CCCTAA) which aids the accuracy of telomere measurement.

4) Did the authors observe any variation in the length of the sub-telomeric regions, which are also known to harbor variation?

Response: Please see discussion of the subtelomeric regions in responses to Reviewers #1 and #2.

5) Did they see any non-canonical telomere repeat variants in their data? This has been reported and would be expected here as well.

Response: Due to the telomere sequence correction function now built into the ONT basecallers, we cannot assess telomere repeat variation. That said, in data produced on earlier R9 flow cells with R9 basecallers (guppy 6.3) we did see degenerate telomere sequences likely resulting from sequencing errors as originally described in Tan, et al. (Reference 42).

6) They used a training-to-test ratio of 7-3 for their binary machine learning classifier, but this will need to be validated on a larger, and independent cohort. Also, any differences and confounding variables should be added to the model and tested, such as the differences by sex, medical status, drugs, or background.

Response: Unfortunately, we cannot obtain this data for the healthy participants (other than sex) as they are anonymous blood donors collected under the Stanford Blood Center's blanket research IRB protocol. Additionally, we believe our sample size is too small to meaningfully parse differences in these epidemiological variables.

7) The code and methods for the classifier are not available for review, and this is required for new methods.

Response: We will make this available for review during this resubmission.

Machine Learning Response Section

We are aggregating the major comments relating to the machine learning model presented in Figure 3 in one section to respond to them in unison.

Reviewer #1

...Therefore, it seems rather redundant for the need to train a machine learning model to do classification for such small sample size (as illustrated in Fig. S8). To demonstrate the usefulness of the machine learning model, perhaps a larger sample size such be utilized to project its application.

Reviewer #2

11) The authors had also mentioned that a “randomized validation set” was used during training (page 15, line 7). Was this resampled from the training/test data? In which case, does this mean that the validation data overlaps with the training data?

12) Figure 3j – It would appear to me that only 14 healthy individuals and 7 individuals with RTEL1 mutations (3 diseased and 4 carriers) were used to develop the machine learning model? This sample size therefore seems too small, and it is therefore not clear how generalizable this model is. When comparing healthy to diseased individuals to carriers, was the comparison done only between 14 healthy vs. 3 carriers? Also, it is not clear to me how the ROC curve was generated from the text. To generate an ROC curve, it is necessary to vary the cutoff value of a “score”. However, it is unclear to me what cutoff value was varied and how it was derived. Overall, I think the results on the machine learning model (which is not as critical for the whole story) in this manuscript needs much further support and substantiation.

Reviewer #3

6) They used a training-to-test ratio of 7-3 for their binary machine learning classifier, but this will need to be validated on a larger, and independent cohort. Also, any differences and confounding variables should be added to the model and tested, such as the differences by sex, medical status, drugs, or background.

7) The code and methods for the classifier are not available for review, and this is required for new methods.

Response: To summarize these comments: the reviewers correctly point out that our neural network model is trained on a very limited dataset of 14 healthy individuals and 8 TBD variant carrier patients, and this can result in an overfitted model whose performance may not generalize to data from a larger, more diverse population. Additionally, the PCA plot (addressed in detail earlier in this response) sufficiently supports the argument that the summary statistics of the telomere length distribution alongside age can distinguish healthy individuals from those with telomere maintenance defects. Finally, the point is made that there could be modifying, non-genetic factors which could impact a predictive model such as the one we have here (and this question itself would necessitate a larger, independent cohort).

These are all valid concerns and suggestions with respect to the model. Our intention in training a binary classification model was to show, as a proof-of-concept, that ensembles of telomere length distributions may be useful and tractable data for downstream clinical applications, many of which rely on calculators or diagnostic tests which accept quantitative clinical data as input and provide an output meant to aid clinicians in the diagnosis or risk assessment of specific disorders. In the area of telomere biology disorders, Flow-FISH of peripheral blood leukocytes can be thought of as one such test as a patient's mean PBL telomere length "scoring" sufficiently low relative to their age group may prompt a clinician to order subsequent targeted exon sequencing to locate a genetic defect of telomere maintenance. Indeed, samples from the Stanford patient in this manuscript (TB32) were brought to us out of concern for a potential telomere biology disorder and short telomeres by digital telomere measurement and flow-FISH prompted the managing clinician to order exon sequencing and ultimately discover the novel TINF2 frameshift mutation we describe in the manuscript. We believe demonstrating that our method can potentially play a similar role as flow-FISH in the management of telomere disorders will be of interest to clinician and clinician-scientist readers of our manuscript and *Nature Communications*.

That said, we agree that the presentation of a neural network model trained on a small dataset without validation in an independent cohort significantly limits the strength of the model as a proof-of-concept of the above. To address this, we have —as one reviewer suggested—trained a much simpler binary classification model by linear regression. Whereas the previous neural network model worked by accepting a linear vector of patient data (age, chromosome-specific telomere lengths, and phenotype), this logistic regression model works on a much more simplified linear vector containing age, the summary statistics of the telomere length distribution (as with the PCA plot: 25th percentile, median, mean, 75th percentile), the phenotype (diseased, carrier, healthy where the distinction between diseased and carrier is made by the empirical presence of clinical symptoms at time of sampling) and then outputting a probability of belonging to group 0 or 1 depending on the model. Additionally, after responding to other reviewer comments relating to the accuracy of chromosome-specific mapping due to the sequence diversity and complexity of the subtelomeric region in humans, we thought it best to exclude chromosome-specific information from model training. The ROC curve for the new linear regression binary classification model is below, once again for three cases originally presented in the manuscript (healthy vs diseased, healthy vs carrier, and healthy vs carrier or diseased).

The previous ROC curve was generated by sliding the threshold for the predicted probability of a group belonging to the diseased / carrier / diseased or carrier condition, and this one similarly reflects the TPR and FPR of the model depending on the threshold set for belonging to the respective phenotype category. Although our data remains limited, we still see that binary classification using telomere length distribution summary statistics and age is a sensitive test for all three phenotypes as all have an AUC > 0.9.

This type of model is less likely to suffer from overfitting. We have also re-emphasized in the text the limitations of a model trained on a small dataset and without validation in a larger, independent cohort. Unfortunately, we believe the time, resources, and expense required to organize, recruit, sample, and then sequence healthy and diseased individuals to be too great to undertake for inclusion in this manuscript. We would like to focus on the introduction of this methodology, its potential, and the other observations and conclusions we make from our data and have edited figure 3 to exclude the binary classification model, and replaced the text discussing the neural network model with text discussing the logistic regression model (and its limitations). The ROC curve above as well as a table summarizing the model results has been added to the supplementary figures of the manuscript. We hope that our work will provide the tools for and inspire others to pursue high-resolution digital telomere measurement at the epidemiological scale in a setting where it is also possible to evaluate modifying non-genetic factors.

The linear regression model and the code used to produce it and the ROC curve will be made available for review.

REVIEWERS' COMMENTS

Reviewer #1 (Remarks to the Author):

See comments to rebuttal

Comments to rebuttal:

Major comments:

1. *The authors indicated that “Telomere-containing reads were identified by aligning telomeric repeats to the chromosomal termini of the recently completed telomere-to-telomere human genome.” Given the extreme heterogeneity of subtelomeric region, use of single haploid genome may not be sufficient. More data should be provided to demonstrate its applicability to resolve the subtelomeric regions.*

Response: We agree that there is significant heterogeneity in the subtelomeric region; however, for the purposes of telomere alignment and measurement, the haploid hs1/CHM13v2.0 reference genome is sufficient. We have included additional analysis below to demonstrate this. Below, we compare telomeres measured by our pipeline (Telometer) with Oxford Nanopore’s proprietary pipeline (TeloSeq) (which we have been able to test via nextflow) using FASTQ reads generated from a HEK293T

SuperTERT, HEK293T GFP or HG002 telomere enriched library. ONT’s pipeline uses a custom diploid reference genome based on the HG002 cell line (from the GIAB project) as opposed to the custom CHM13v2.0/hs1 + Stong 2014 subtelomere assembly reference genome used by all other Telometer experiments. To compare the two assemblies, we applied Telometer to reads aligned to the custom HG002 reference genome ONT uses in TeloSeq (it is publicly available on the TeloSeq github page <https://github.com/rob234king/Teloseq>). We test all combinations of TeloSeq or Telometer using both reference genomes and demonstrate that there is very strong agreement between the telomere length distributions generated by all combinations of pipelines and reference genomes across all cell lines. TeloSeq for HEK293T GFP is not shown as these data were generated using an R9 flow cell and TeloSeq has only been optimized to work for R10 telomere base calls.

Pipeline	First Quartile	Median	Mean	Third Quartile	Reference	Cell Line	Total Telomere
Telometer	3,027.25	4,074.00	4,270.65	5,256.75	CHM13v2.0 + Stong 2014 (Haploid)	HEK293T SuperTERT	21,402.00
TeloSeq	3,147.00	4,068.00	4,220.48	5,138.00	HG002 (Diploid)	HEK293T SuperTERT	2,641.00
Telometer	3,073.00	4,141.00	4,238.21	5,317.00	HG002 (Diploid)	HEK293T SuperTERT	13,105.00
Telometer	4,125.00	5,039.00	5,115.26	6,169.75	HG002 (Diploid)	HG002	7,040.00
Telometer	4,209.00	5,064.00	5,307.37	6,110.00	CHM13v2.0 + Stong 2014 (Haploid)	HG002	11,825.00
TeloSeq	4,292.00	5,123.00	5,294.49	6,049.00	HG002 (Diploid)	HG002	1,837.00

The reference genome and pipeline used can affect the number of telomeres measured but does not affect the telomere measurement distribution (presumably as long as there are sufficient telomere measurements, and in this case all examples are well above the point of diminishing returns for the standard error of the mean, as depicted in Figure 1). In general, it appears that using a more inclusive reference genome (like CHM13v2.0 + Stong 2014) leads to more telomere measurements, despite being haploid.

2. Detail of the telomere capture procedure should be described.

Response: The protocol for telomere capture library preparation used to generate the data in this paper is detailed in the Methods section, copied below. Additionally, the github page for Telometer (<https://github.com/santiago-es/Telometer>) includes an annotated, bench-friendly version of the protocol which will be updated periodically as the protocol evolves and improves. From our Methods section:

Telomere capture sequencing nanopore library preparation

Barcoded telomere capture oligos were annealed to sequencing tether (seqTether) by mixing equimolar amounts of both oligos in low TE buffer, heating to 95°C for 2 minutes and then being allowed to cool at room temperature for an

hour. Approximately 3 ug of HMW genomic DNA was ligated to barcoded, freshly duplexed oligos in a 100 µL ligation reaction (10 µL 10X rCutSmart buffer, 5 µL 5 µM duplex capture oligos, 2 µL 2000U/µL T4 DNA ligase, 1 µL 10 mM ATP, 3 µg gDNA, nuclease-free H₂O up to 100 µL) overnight at 37°C. The following day the ligation reaction was heat inactivated at 65°C for 10 minutes. Potential gaps between the capture oligo and the double-strand/single-strand junction were then filled in using the same reaction tube by adding 2 µL (4U) *Sulfolobus* DNA Polymerase IV (NEB #M0327S), 12 µL 10X ThermoPol Buffer, 1 µL 20 mM dNTPs, 1 µL 10 mM ATP, and 4 µL of nuclease-free H₂O followed by incubation at 56°C for 2 minutes and then 72°C for 15 minutes with shaking (500 RPM). Promega ProNex size selection beads were then added to the ligation reaction at a bead-to-solution ratio of 1.6 and the solution was equilibrated on a rotating mixer for 5 minutes at room temperature. The bead solution was then pelleted on a magnet, washed twice with 80% ethanol, allowed to dry on the magnet for 3 minutes, and then eluted in 60 µL of ONT elution buffer for 15 minutes at 37°C. The complete volume of eluted capture-oligo ligated DNA was then ligated onto the ONT sequencing adapter (AMII for R9, NA for R10 libraries) per ONT's barcode ligation sequencing protocol (5 µL sequencing adapter mix, 20 µL 5X Quick T4 DNA Ligase Buffer, 10 µL Quick T4 DNA Ligase, 5 µL nuclease-free H₂O) at room temperature for one hour. The rest of the library preparation and sequencing protocol is performed as for whole-genome sequencing above.

Comment: Based on the author's description, there is no purification step for telomere-containing genomic DNA fragments. The only enhancement is the preferential capture of the telomere-containing reads through the preferential ligation of ONT adaptor to telomeric single-stranded overhangs. While the authors have claimed "resulted in enrichment for telomeric reads by several thousand-fold without significantly impacting the measurement of the telomere length distribution". No data is shown for the estimation of the estimated enrichment.

3. *The TRF data for HEK 293T should be included in Fig. 1 in order to compare the telomere length estimation between TRF and digital mean telomere length estimation.*

Response: The manuscript already describes direct comparisons using the same genomic DNA preparation in our 14 healthy individuals and the two samples from the peripheral blood and bone marrow preparations for patient TB32 (TB32-B and TB32-M, respectively) in figure S1. These DNA were treated side-by-side using the same amount of input DNA (10 µg) with the same restriction enzyme cocktail (HinfI / RsaI) for the same duration of genomic digestion (~16h) since TRF results can vary significantly due to these factors. It is this TRF that the comparison figure (Fig. 1d-e) is based on.

We have since performed TRF on the HEK293T genomic DNA in Figure 1 as described above and include the figure alongside this response. In comparing this TRF with the digital telomere measurements already reported in the manuscript, we similarly observe that TRF overestimates

the mean telomere length by approximately 2 kb, in agreement with our other results.

Comment: The original Southern blot with radioactively-labelled DNA ladder should be used.

4. Given the heterogenous length of telomeres, it is surprising that the SE of measurement reach a maximal precision of 30-40 base pairs using just 1000 telomere containing reads (Fig. 1h). How many independent replicates in different nanopore flow cell have been sequenced? For the Bootstrapping analysis, how many total reads were used for simulation? Is the Bootstrap resampling analysis performed with replacement or without replacement?

Response: We have found that our method is highly reproducible across separate sequencing runs. To give one example, the top left panel of the figure below shows telomere length distributions from HEK293T cells transiently transfected with GFP from the manuscript's original sequencing run and from a new sequencing library prepared from the same genomic DNA extract as the original sequencing run produced for this revision of our manuscript. The summary statistics (top right panel) show strong agreement between both sets of measurements.

The bootstrapping analysis in the manuscript (Fig. 1H) was performed with replacement using a dataset containing ~1300 telomeres. The result described is reproducible using any suitably large dataset with the same bootstrapping conditions. To demonstrate this, we have rerun the bootstrapping analysis using a set of over 21,000 telomeres from an HG002 telomere capture sequencing experiment and similarly observe an inflection point for the standard error of the mean of the telomere length distribution near 250 telomere measurements and a plateau near 1000 telomeres which decreases very slowly with additional telomere measurements. The bottom right panel shows the entire curve produced by the bootstrapping analysis while the bottom left zooms in to facilitate comparison to Fig. 1H.

Experiment	25th Percentile	Median	Mean	75th Percentile	Total Telomeres
Original	2,797.00	3,601.00	3,816.47	4,746.00	1,066.00
Revision	2,678.00	3,612.00	3,791.65	4,764.00	14,223.00

Finally, we collaborated with the Karlseder lab to sequence independent samples of HG002 cells sourced from the Coriell institute to compare our techniques on the same cell line. In the Karlseder lab's manuscript co-submitted alongside ours, the summary results are below:

Manuscript	25th Percentile	Median	Mean	75th Percentile
Schmidt, et al. (Karlseder, TeloSeq)	4011	4979	5247	6115
Sanchez, et al. (Artandi, Telometer)	4209	5064	5307	6110

We believe this demonstrates the technical reproducibility of telomere measurement by long-read sequencing even when using distinct sets of telomere capture oligos, distinct analysis pipelines, and in orthogonal laboratory conditions with different operators to a much greater degree than any other telomere measurement method to date.

Comment: The authors claim "Bootstrapping analysis of our measurement results suggests the standard error of measurement by our method decays exponentially with additional telomere measurements eventually resulting in a maximal precision of 30-40 base pairs". However, the maximum number of reads used for Bootstrapping analysis is only 1500bp. If higher read depth can be achieved, will this affect the accuracy? These information will be useful to the readers.

5. *The human ESCs are known to have telomere length round 10-14kb. However, in Fig. 2a, the ESCs with WT TIN2 only have a mean telomere length around 5kb? Similar results are shown for Fig. 1j. Can the author confirm the results are generated from ESCs with WT TIN2? Please use the ES cells such as WA01 and WA09 as control. If this is the case, can the author explain such discrepancy.*

Response: As the reviewer mentions, individual human embryonic stem cell cultures have an intrinsic telomere length "set point" that is maintained during extended culture as a stable average telomere length. However, there is variation in mean telomere length between ES cell cultures. The cells studied here are derived from WIBR3 hES cells, with subclones maintaining a stable telomere length ~7kb as measured by TRF (following overnight Mbol/AluI digestion) for 150 days in culture (Choo et al 2022, Reference 31 in the manuscript). As in Figure 1 panels I and J, we consistently find that TRF measurements of telomere length overestimate mean telomere length by approximately 2-3kb (when digested overnight with HinfI/RsaI), likely due to undigested subtelomeric regions which are retained during restriction digest of the end-terminal chromosomal fragments. Together, this demonstrates the telomere length data for this hES culture is internally consistent.

Below, we have included Sanger genotyping of the DC cluster to demonstrate the targeted mutations at the TIN2 locus in the cells sequenced for this manuscript:

Regardless of parental cell line, the purpose of this experiment was to show that long-read sequencing of telomeres robustly detects the altered telomere length set point of an isogenic stem cell line with a mutation in the TIN2 DC cluster. A previous publication has demonstrated that this mutation produces approximately a 1-1.5kb reduction in telomere length which is stable over time (Choo et al 2022). This is recapitulated in our long-read sequencing data.

Similarly, the hESC WT cells shown in Fig. 1J are from the WA01 background and the accompanying TRF (performed with overnight HinfI/RsaI digestion) are presented side by side. Our laboratory has heavily utilized this cell line over many passages for a variety of experiments and it is possible that over that time changes have occurred which altered the telomere length of the wild-type cell line, but it is the appropriate control as the parental line of the hESC PARN KO cells which are presented alongside it in both the digital and analog telomere measurement experiments with the goal of recapitulating the degree of telomere shortening observed in individuals with genetic PARN defects. It is likely that other ES cell populations will have differing telomere lengths but reporting on such variation is outside the purpose of this study.

Comment: The telomere length presented in Figure 1J have a mean telomere length of around 5-6kb. If indeed this is from WA01 background, this is in drastic

contrast to the known ESC cells' telomere length set point (including WA01), which is about 10-14kb. This also echoes the authors' observation that "we consistently find that TRF measurements of telomere length overestimate mean telomere length by approximately 2-3kb". This seems to be drastic contrast to recent results published in bioRxiv suggesting that the mean telomere length measured using ONT is in agreement with telomere length on Southern blot {Karimian, 2024 #2138}. These discordances highlight further improvement may be necessary to achieve accurate telomere length measurement using ONT, especially when different version of ONT flow-cells were used that may bring difficulties to compare data generated from different labs.

Also, HinfI/RsaI digestion will generate very short subtelomeric regions that are <500bp. This is far from the authors' estimated 2-3kb differences.

6. For Fig. 2c, increased short telomeres correlated with days post TERT knockout is understandable and expected. However, why there are more short telomeres (0-1000) in WT than TIN2 284R/+ and TIN2 284R/284R; likewise, why TIN2 284R/+ cells have more extreme short telomeres (0-1000) than TIN2 284R/284R...This is very confusing as the data also show that the distribution of shorter telomeres (1000-2000) in TIN2 284R/284R>TIN2 284R/+>WT.

Response: Thank you for this observation. We also noticed this phenomenon, but do not yet understand the biological basis for this observation so did not comment explicitly in the text. We are interested in pursuing the nature of this phenotype in the future and believe it could be related to the fact that unlike genotypes which specifically impair the telomerase enzyme, the TIN2 284R mutation operates through a poorly understood mechanism. We included this experiment to illustrate two points: first, that as with PARN KO, we can recapitulate the telomere phenotype of a known DC genotype; second, that we can see progressive shortening in a manner that is dependent on the dosage of the mutant allele which could not be previously appreciated with TRF alone (Choo, et al. 2022, Reference 31). It will be interesting to continue to explore the telomere biology of TIN2 and shelterin using longread sequencing, but we believe that doing so is outside of the scope of this manuscript introducing the capabilities of the method for the first time.

Similar results were shown in Fig. 3h, younger individual seem to have more telomeres that are (0-1000) in length, while less telomeres that are (1000-2000) and so on. These results seem to contradict each other. This is unexpected and please elaborate more on this.

Response: We do not believe that the difference in the 0-1000 bp fraction between the various healthy donor groups is significant and the difference that is being observed is likely due to the small population studied in this manuscript. The linear regressions of the various telomere length summary statistics versus age in this cross-sectional cohort also demonstrate that the relationship between age and length of shorter telomeres is less steep than it is for longer telomeres. Additionally, previous epidemiological studies of telomere length in much larger cohorts have observed significant variation between healthy

individuals, which we also observe in this small cohort. The power of our method we are attempting to highlight here is that statistically significant relationships can still be gleaned from the data despite the small sample size (as illustrated in Fig 3E-F) and that the slope of the correlation between age and telomere length varies for different summary statistics of the telomere length distribution.

Comment: As the authors have indicated that the contradicting data remain unresolved. Can the authors provide a reference to the claim “The linear regressions of the various telomere length summary statistics versus age in this cross-sectional cohort also demonstrate that the relationship between age and length of shorter telomeres is less steep than it is for longer telomeres.”

7. For Fig. 2i, can the author indicate whether the graph shows the p arm or q arm for each chromosome? Why not both chromosomal ends? It is known the telomere length of two alleles from the same chromosomal end can differ dramatically. Can the author provide the results in allele specific manner to support their conclusion?

Response: This figure originally aggregated both arms. Given that it is true there can be variation at the level of individual chromosome arms, we have addressed this in the response to the next comment.

Comment: Please provide the reads depth for each chromosomal end.

8. The authors indicated that “Figure 2 results indicated that chromosomes with shorter telomeres in the control group (GFP) showed a greater magnitude of elongation relative to chromosomes with longer telomeres when treated with telomerase overexpression (Fig. 2h).”

This conclusion maybe true when you only compare Chr. 3 to Chr. 10. However, one can obviously see that: while Chr. 7 has longer telomere length than Chr. 20 in cells transfected with GFP; the telomere elongation for Chr. 7 is obviously more dramatic than Chr. 20 in cells transfected with hTERT and hTR/TSQ1. In addition, there seem to be a lot of discrepancy in the telomere elongation upon hTERT and hTR/TSQ1 overexpression, as the mean telomere length for Chr. 16, 17, and 19 in cells transfected with hTERT and hTR/TSQ1 seem to be shorter than those in cells transfected with GFP control. These discrepancies could be cause by the lack of sequencing depth? Can the author indicate the read number for each chromosome on the graph?

Response: As discussed in response to major comment number 4, the original HEK293T GFP data had only around ~1000 telomere reads, which at the level of specific chromosomes translated to very low coverage in the GFP sample of certain chromosomes, and as discussed the aberrant HEK293T genome means

the chromosome and chromosome arm dosage / cell is not as is expected for a normal diploid genome. We did not take this into consideration when originally presenting these data. To correct this, we have re-sequenced a new extraction of HEK293T GFP DNA from live cell suspensions previously aliquoted at the collection timepoint of this experiment which were in long-term liquid nitrogen storage, obtaining over ten-thousand more telomere reads. The raw counts per chromosome and chromosome arm are shown below. As is apparent from the data, the counts per chromosome and chromosome arm are not uniform for this aneuploid genome, but they are in similar proportion for each chromosome arm across both the original SuperTERT data (which already contained tens of thousands of telomeres) and the revised HEK293T GFP data. We have also included a new figure showing the boxplots for each chromosome (this time also separated by arm).

While we still believe there is evidence of a trend for shorter chromosome arms to experience more elongation, we take the reviewer's point that there could be additional complexities involved in illustrating this hypothesis definitively which fall outside the intended scope of this manuscript. We have amended the text in the relevant section to read:

"We observed variability in the amount of telomere elongation experienced by different chromosome arms after transient telomerase overexpression (Fig. 2I)." And edited this discussion to highlight that our data in healthy human aging also supports a model of preferential telomerase activity at the shortest telomeres, as other studies have proposed.

Transfection Plasmids GFP hTERT + hTR/TSQ1

Comment: As commented above, the read depth may be the reason of such heterogeneity of telomere length elongation.

9. *The authors indicated that “Quantification of the fraction of telomeres of various sizes in each aging cohort shows that the fraction of the distribution comprised of shorter telomeres increases with age as the longer telomere fraction shrinks with the notable exception of the two shortest fractions which appear to be relatively stable with age in PBLs (Fig. 3h).”*

Can the authors indicated which two shortest fractions which appear to be relatively stable with age in PBLs?

Response: The 0-1000 and 1000-2000 bp fractions. The text was edited to clarify this point.

Comment: Here, I would like to quote the authors' comment from their rebuttal for point 6 “We do not believe that the difference in the 0-1000 bp fraction between the various healthy donor groups is significant and the difference that is being observed is likely due to the small population studied in this manuscript.” Are these difference consistent and informative?

10. *The authors indicated that “this method is high throughput.”*

How many samples have been pooled into one flow cell for sequencing? Please also explain the methodology in detail.

Response: In the experiments contained in this manuscript we have successfully pooled up to 8 samples on a single PromethION flow cell. The methodology for telomere capture is detailed in the Methods section “Telomere capture sequencing nanopore library preparation”, and language has been added to explain barcode pooling:

For barcoded, pooled libraries up to 8 samples were pooled prior to sequencing adapter ligation at equal molecular weights up to a total of 1000 ng. Sequencing adapter ligation and subsequent preparation for flow cell loading is as previously described.

Comment: Based on the author's description and rebuttal to point 2, no enrichment of telomere-containing genomic DNA fragments was performed. Given the telomeric content of human diploid cells is only 0.015% of the total genome the vast human genome, an enrichment step seem to be necessary for telomere length measurement using long-read sequencing platform. How many reads per sample can be obtained? The method described seems to be inconsistent throughout. While the authors indicated that “up to a total of 1000 ng” was used prior to sequencing adaptor ligation here, they also indicated that “at least 3 ug of genomic DNA should be used” on their response for point 11. Can the authors be clear on the methodology?

11. *The authors indicated that “Telomere measurements made by long-read sequencing are comparatively much richer than TRF, flow-FISH while requiring less input DNA than either existing technique and are not limited*

*by polymerase processivity like PCR-based methods STELA and TeSLA.”
How much input DNA is used for Oxford Nanopore sequencing in this
manuscript? What is the expected output of telomeric reads with that input?*

Response: At least 3 µg of genomic DNA should be used as input for telomere capture libraries, while WGS libraries can be prepared with as little as 1 µg of genomic DNA. With respect to telomere capture, the degree of telomere enrichment and expected output depends on the input used, as has been described by ONT in their TeloSeq release:

[redacted]

For the telomere capture libraries represented in this manuscript, we routinely and consistently used 5 µg of DNA as input, and the relative yield of telomere measurements per gigabases sequence varied as illustrated in Fig. 1C.

Comment: While it is true that the long-read sequencing platforms (ONT and PacBio) can achieve better accuracy than TRF, flow-FISH, STELA and TeSLA, I do believe the current requirement of genomic DNA input for long-read sequencing platforms is much higher than these aforementioned methods. PCR based methods such as STELA requires <100ng of genomic DNA.

12. For Fig. 3g, please provide detailed analysis steps including the underlying raw data matrix used to generate the PCA plot, and please provide detailed explanation for this figure in the main text. It is obvious that the first two PCA components are sufficient to differentiate the 3 subgroups. Therefore, it seems rather redundant for the need to train a machine learning model to do classification for such small sample size (as illustrated in Fig. S8). To demonstrate the usefulness of the machine learning model, perhaps a larger sample size such be utilized to project its application.

Response: The raw data matrix used as input for the PCA is reproduced below. The summary statistics of the telomere length distribution and age were used as input.

sample	mean	first_quartile	median	third_quartile	Age
DC1	3,088.41	1,163.25	3,012.50	4,389.00	50
DC2	3,040.89	1,554.00	2,618.00	4,199.00	45
DC3	3,661.53	2,537.50	3,597.00	4,751.00	36
DC4	4,563.35	3,054.00	4,490.50	5,729.00	35
DC5	4,381.68	3,383.00	4,530.00	5,328.00	6
DC6	4,489.95	3,539.00	4,099.00	5,417.00	21
DC7	3,751.70	2,931.00	3,352.00	4,675.00	47
TB32-B	2,885.06	2,021.00	2,940.00	3,596.00	60
TB32-M	3,263.86	2,322.50	3,054.00	4,043.50	60
ma01	4,931.01	3,643.00	4,622.00	6,101.00	46
ma04	4,593.73	3,369.00	4,352.00	5,381.00	65
ma06	5,334.80	3,884.00	5,098.00	6,254.00	50
ma13	4,862.85	3,609.00	4,634.00	5,863.00	46
ma20	4,788.37	3,439.50	4,271.00	5,721.00	48
ma66	5,249.87	3,909.00	5,072.00	6,483.50	35
oa51	4,496.32	3,645.50	4,086.00	5,511.00	73
oa60	4,637.12	3,427.00	4,357.00	5,729.50	77
oa68	4,302.34	3,116.50	4,155.00	5,284.00	75
ya40	4,715.79	2,747.75	4,393.00	6,474.00	20

ya41	6,371.66	4,270.50	6,706.00	7,922.00	18
ya45	7,421.13	5,413.00	7,586.00	9,164.25	18
ya55	5,156.13	3,502.00	5,064.00	6,630.75	19
ya56	5,954.76	4,417.00	6,080.00	7,580.75	20

The code (R) used to produce the PCA analysis is reproduced below.
 Telometer_data stands in for a telometer output file, dplyr is used to produce the summary statistics of the column "telomere_length", and then prcomp is used to perform PCA.

```

—
statistics <-
telometer_data %>
%
group_by(sample)
%>%
dplyr::summarise(
  mean = mean(telomere_length, na.rm
= TRUE),  first_quartile =
quantile(telomere_length, 0.25, na.rm =
TRUE),  median =
median(telomere_length, na.rm = TRUE),
  third_quartile = quantile(telomere_length,
0.75, na.rm = TRUE),  Age = first(Age)
)
sample_conditions <- distinct(df, sample, .keep_all = TRUE)[, c("sample",
"Phenotype", "Age")]

# PCA with prcomp()
statistics_data <- select(statistics, -sample) # remove sample name
so it is not used in PCA
pca_result <- prcomp(statistics_data, center
= TRUE, scale. = TRUE)
pca_data <- as.data.frame(pca_result$x[,
1:2])
names(pca_data) <- c("PC2", "PC1")
# Add back in the sample data, aligning with PCA
results_pca_data$sample <- statistics$sample
pca_data <- merge(pca_data, sample_conditions, by = "sample")

# Plot (ggplot2)
p <- ggplot(pca_data, aes(x = PC2, y = PC1, color = Age, shape =
Phenotype)) + geom_point(size = 3) +
  scale_color_viridis_c(option = "viridis", name = "Age") +

```

```
scale_shape_manual(values = c("Healthy" = 16, "Carrier" = 17, "Diseased"
= 18)) + geom_text(data = subset(pca_data, Phenotype != "Healthy"),
aes(x = PC2, y = PC1, label = sample, color = Age), vjust = -0.5, hjust =
0.5, check_overlap = TRUE) +
labs(title = "", x = "PC2", y = "PC1") +
theme_minimal() + theme(text=element_text(size=16))
```

We have also amended the text to discuss the PCA directly.

Comment: Please include the telomere-containing reads number in the table. In addition, the newly provided excel table (file name: 462782_1_data_set_8811944_s11nf9.csv) lacks clear column descriptions, making it difficult to interpret the data accurately. For instance, while HG002 shows 10465 telomere reads with a coverage of 3.3, DC1 has only 134 telomere reads but with a higher coverage of 7.7.

13. *On page 5, the authors described that “In RTEL1 patients and carriers, we no longer observe a difference in the extent of decreasing telomere length in the first and third telomere length quartiles with age (Fig. 3b)”*

However, Fig. 3f shows that there is a decreasing trend of first and third telomere length quartiles with age. Can authors please explain the discrepancy shown between Fig. 3b and 3f?

Response: The original text contains a typo here and the quoted text should refer to Fig. 3E. We also realize the original text as written is unintuitive. Our intention was to describe the differences in the slope of the relationship between cross-sectional Age and the various telomere length summary statistics in healthy individuals or RTEL1 variant patients. We point this out immediately after highlighting that TERT null hESCs experience rapid first quartile telomere length shortening as well as third quartile telomere length shortening while white blood cells from healthy individuals experience a “slower” decrease in the first quartile telomere length relative to the third quartile. We have edited the text to reflect this.

Additionally, in response to a different reviewer’s comment Figures 3E and 3F will be combined and TB32 will now be included in the DC cohort as the patient has now been confirmed to fit the clinical diagnosis of a telomere biology disorder.

14. *In Fig. 3b, DC1 and DC2 are two samples collected from same individual with 5 years apart. While it is found that the fraction of smallest telomere (0-1kb) is dramatically increased from 10% in DC2 to 20% in DC1, the median telomere length of DC1 (age 50) seems to be longer than DC2 (age 45). This seems to*

contradict the ability of digital telomere measurement to detect telomere shortening with age as indicated by Fig. 3a.

Response: We disagree that these data are contradictory. It is important to remember that telomere biology disorder patients are not experiencing healthy aging and illustrating the distinction between healthy aging and telomere biology disorders as observed through the quantitative lens of previously inaccessible telomere length distributions is a major contribution of this manuscript. Finally, we believe that the increase in short telomeres over this five-year period is consistent with progression of this individual's disease with age.

That said, we believe that there may be interesting clonal dynamics in this patient's hematopoietic system which could be contributing to this observation, including entering levels of telomere shortening in which negative selection due to telomere shortening mediated senescence is creating a bottleneck for the hematopoietic stem cell pool, and we are continuing to explore these questions in ongoing projects in our laboratory. While the following data will not be included in this manuscript as the relationship between telomere length and clonality is not a focus of this paper, we used cuteSV to characterize structural variants in samples DC2 and DC1 and observed a significant increase in structural variants over a 5-year period. We additionally performed targeted sequencing of a panel of genes commonly involved in the development of clonal hematopoiesis and identified a significant STAT3 mutant clone (VAF ~14%) at age 45 which was no longer present at age 50, suggesting an underlying process of stem cell competition or selection occurring in this patient's marrow. In the figures below, S1 corresponds to DC2 and S2 to DC1. We are interested in continuing to explore these questions with long-read sequencing.

Targeted Sequencing of CH Genes

Sample	Gene	VAF	Mutation Type
S2	TET2	0.99	germline
S2	TP53	0.52	germline
S1	TET2	0.99	germline
S1	TP53	0.53	germline
S1	STAT3	0.14	clone

Comment: Please provide the read depth of telomere reads for each samples shown. This is important for the readers to understand whether these inconsistencies are due to read depth. While it is true the current ONT method provides better accuracy to telomere length measurement, such accuracy is depended on the read depth as the authors have shown in their bootstrapping analysis.

15. In Fig. S3 (bottom plot), the spread of subtelomere length seems too big. Is that because of low sequencing depth? If so, please refine the length estimates by including more reads following the suggested depth in the simulation study (Fig. 1h).

Response: The purpose of this supplementary figure is to demonstrate that the length of remaining subtelomere on telomeric reads is dependent on the number of restriction enzymes used during genomic digestion, which is itself not particularly surprising or interesting. However, given that TRF and flow-FISH are most likely overestimating telomere length due to undigested subtelomeric DNA that is included on the terminal telomere fragment, we thought it was important to document this quantitatively using the data we've already generated. The single enzyme EcoRV digestion represented in the bottom of S3 was performed for only 30 minutes - 1 hour, as detailed in the methods section pertaining to telomere capture library preparation, so it is also likely incomplete.

That said, we've refined the estimates by using subtelomere length measurements from an experiment with >21,000 telomeric reads instead of the original ~2,000 reads used in the original S3 EcoRV figure.

Reviewer #2 (Remarks to the Author):

Summary

=====

Sanchez et al had addressed most of the points I raised. Most significantly, they had replaced their neural network model with a regression model. Additionally, they have made the presentation of their data much clearer. The following are points that I had previously raised, but are not fully addressed yet:

Major points

=====

4) As before, I still have significant concerns about the misassignment of the telomeric reads to the wrong chromosomal arms. As also highlighted by the authors, the reliability of mapping of true telomeric reads to the correct chromosome arm will also depend on the quality of the reference genome. Thus, in the absence of a complete reference genome of HEK293T, mis-mapping of telomeric reads between chromosomal arms is likely to occur.

I would therefore suggest that the authors (i) provide further evidence to show that telomeric reads are mapped to the correct arms, (ii) only highlight the chromosomal arms that can be reliably assigned, or (iii) to remove this panel (Figure 2i) from this figure as it is not as pertinent to their story.

7a) The statistics that I had requested should be quite easy for the authors to collect. However, only the number of telomeric reads were presented. It is unclear to me how many reads were generated in total. Again, these metrics would be very useful for readers who are interested in understanding the relative performance of different approaches for telomere enrichment.

7b) These numbers should be presented as part of the table for point 7a. Here, it is also quite surprising to me that most of the reads were observed from the C strand while in the reply to question 3 by Reviewer #3, the authors had observed more reads with the AGGGTT motif (i.e. G strand) than with the AACCT motif (i.e. C strand). The two pieces of data that the authors had presented therefore seem to contradict each other?

8) The authors had stated in their revised text that:

"including more rapid shortening of the third quartile relative to the first (but this trend does not reach statistical significance)."

If the result does not reach statistical significance, there is insufficient evidence for the authors to conclude that the third quartile is shortening more rapidly than the first. I would therefore encourage the authors to refrain from making such a claim.

Minor

=====

9) If the comparison is between the author's approach with that from citation 26, this should be made very clear. The authors had reported that this difference is due to the sequencing platform in the legend of Figure S3. However, these differences could instead have occurred due to how these libraries were prepared rather than due to the sequencing platforms.

Reviewer #3 (Remarks to the Author):

Overall the paper is very improved, and they have also addressed many of the questions about k-mer differences and mapping.

However, a request for the code and methods for the classifier was not completed, as far as I can tell? It just says "We will make this available for review during this resubmission."

We have now addressed all the reviewers queries and I would like to thank each reviewer for their thoughtful and helpful comments. To summarize, we have:

- Added additional information to the sequencing metadata supplementary spreadsheet
- Included a second supplementary spreadsheet containing the raw data used to generate the figures in the manuscript
- Edited the text to respond to comments made by reviewers below, including removing panel 2i from Figure 2 to address concerns about the accuracy of chromosome-specific mapping for HEK293T sequencing reads.
- Included a figure showing the HG002 chromosome, allele-specific telomere length distributions and edited the text to explain the methodology and results.
- Sequenced hESC TIN2 heterozygous and homozygous 284R mutants more deeply to bolster the observation made on the original submission
- Added a supplementary figure showing the TRF Southern blot for HEK293T genomic DNA (now Fig S1)

Reviewer #1

1. *Comment: Based on the author's description, there is no purification step for telomere-containing genomic DNA fragments. The only enhancement is the preferential capture of the telomere-containing reads through the preferential ligation of ONT adaptor to telomeric single-stranded overhangs. While the authors have claimed "resulted in enrichment for telomeric reads by several thousand-fold without significantly impacting the measurement of the telomere length distribution". No data is shown for the estimation of the estimated enrichment.*

Response: Figure 1C compares the number of telomere measurements per gigabases sequenced between WGS (median ~15 telomeres / Gb sequenced) and telomere capture (median ~5000 telomeres / Gb sequenced) long-read sequencing. This figure shows a relative enrichment of several thousand-fold, on average. Figure 1C was generated using the metadata from all of the experiments included in the manuscript (n=37 total samples, n=14 for telomere capture, n=23 for WGS; please see SeqRunStats supplementary data).

2. *Comment: The original Southern blot with radioactively-labelled DNA ladder should be used.*

Response: We do not routinely use radioactively-labeled DNA ladder for our TRF southern blots. Instead, we run a DNA ladder alongside our sample and capture an ethidium bromide image of the gel with a UV-transparent ruler placed on top of the gel and determine the relative position of the radiolabeled TRF signal against the DNA ladder. The full method used is described in the Methods section, reproduced below with relevant sections highlighted:

"Approximately 4 µg of genomic DNA was prepared in a 50 µL total volume restriction digest solution (1X Fast Digest Buffer, 3 µL HinfI, and 3 µL RsaI) and allowed to digest at 37°C overnight. In the morning, 1 µL each of HinfI and RsaI was added to each digestion reaction and allowed to incubate at 37°C for a further three hours. A 1% TAE agarose gel was prepared and 3 µL per sample underwent gel electrophoresis (125V, 70 minutes) to confirm restriction digest completed successfully (Figure S2). A 0.8% TBE agarose gel was then prepared in a 20x27 cm casting tray after adding 15 µL ethidium bromide to the agarose solution. The entire volume of each restriction digest reaction was then loaded into each well with 1X NEB nucleic acid loading dye in addition to NEB 1 kb reference ladder and gel electrophoresis was performed (85V for 16 hours). In the morning, the gel was dried using a BioRad gel dryer (1 hour under vacuum then 1 hour under vacuum and 50°C). A **UV-translucent ruler was then overlaid on the dried gel before imaging with UV transillumination to establish reference distances for the ladder markers from the well positions.** The dried gel was then incubated in a denaturing buffer (1.5M NaCl, 0.5M NaOH) for one hour with gentle shaking. The denatured gel was washed with deionized water twice before a second one-hour incubation in neutralizing buffer (1.5M NaCl, 1M Tris-HCl, pH 7.4) with gentle shaking. The

neutralizing buffer was decanted and the neutralized gel was washed twice with deionized water. The gel was then rolled vertically into a glass hybridization tube (Thermo-Fischer Scientific) and incubated with pre-warmed hybridization buffer (Invitrogen #AM8670) at 42°C for 30 minutes with rotation. 0.5 μM of $\gamma\text{-}^{32}\text{P}$ labeled telomere probe was then added to the hybridization buffer tube and incubated at 42°C overnight with rotation. The gel was washed once with 2X SSC buffer and twice more with 1X SSC buffer (0.15M NaCl, 15 mM sodium citrate) before exposing onto a phosphor screen inside a lead exposure cassette for 24 hours. Following exposure, the phosphor screen was imaged on a Typhoon scanner.

Both the southern blot images and the ethidium bromide reference ladder were loaded onto ImageJ and aligned. The signal intensities at each position coordinate starting from the bottom of the well in the southern blot image were obtained with the ImageJ line and Measure tools after drawing a line from the bottom of the well to the bottom of the gel through each sample lane.”

We have now added the HEK293T Southern Blot to the Supplementary Figures, as Fig. S1.

3. *Comment: The authors claim “Bootstrapping analysis of our measurement results suggests the standard error of measurement by our method decays exponentially with additional telomere measurements eventually resulting in a maximal precision of 30-40 base pairs”. However, the maximum number of reads used for Bootstrapping analysis is only 1500bp. If higher read depth can be achieved, will this affect the accuracy? These information will be useful to the readers*

Response: Although “1500 bp” is written in the Reviewer’s query, we take this comment to mean only 1500 measurements were used for bootstrapping analysis given the reference to higher read depths affecting accuracy. This is not correct. The figure we provided in the rebuttal comments is a repetition of the bootstrapping analysis using an experiment where over 20,000 individual telomere measurements were made, reproducing the results of the original analysis (a plateau in the standard error of the mean near 30-40bp, with significant diminishing returns in the precision of the mean telomere length estimation near 200 measurements). The bootstrapping analysis provided in the first rebuttal is reproduced below. The figure on the left is only a cropped version of the figure on the right to demonstrate the view provided in the bootstrapping figure in Figure 1.

4. *Comment: The telomere length presented in Figure 1J have a mean telomere length of around 5-6kb. If indeed this is from WA01 background, this is in drastic contrast to the known ESC cells’ telomere length set point (including WA01), which is about 10-14kb. This also echoes the authors’ observation that “we consistently find that TRF measurements of telomere length overestimate mean telomere length by approximately 2-3kb”. This seems to be drastic contrast to recent results published in bioRxiv suggesting that the mean telomere length measured using ONT is in agreement with telomere length on Southern blot {Karimian, 2024 #2138}. These discordances highlight further improvement may be necessary to achieve accurate telomere length measurement using ONT, especially when different version of ONT flow-cells were used that may bring difficulties to compare data generated from different labs. Also, HinfI/RsaI digestion will generate very short subtelomeric regions that are <500bp. This is far from the authors’ estimated 2-3kb differences.*

Response: This is a point raised again by this reviewer from the first round of review. As was discussed in the previous rebuttal, the hESCs referenced are a subclone from a WA01 background which were used as the parental wild-type to derive the hESC TIN2 284R mutants by the Hockemeyer lab. This subclone has already been recorded to have a mean telomere length ~7 kb by TRF following overnight Mbol/Alul digestion which was stable for at least 150 days in culture, and these data were published by the Hockemeyer lab in Choo et al, 2022 (31). As with all other comparisons made between digital telomere measurement using our bioinformatic pipeline and TRF Southern blot, the mean telomere length by TRF is ~2 kb longer than that measured by long-read sequencing.

In reference to Karimian 2024, the final version of this manuscript published recently does not contain a quantitative comparison of TRF Southern blot and digital telomere measurement, so we do not believe we can confidently evaluate the claim that there is no difference between the mean determined by TRF Southern blot and their bioinformatic pipeline, which differs from ours in several potentially relevant ways. The work of comparing various telomere measurement pipelines by long-read sequencing is important, but falls outside the scope of this manuscript. Our manuscript, however, does contain direct quantitative comparison of telomere measurement by long-read sequencing using our approach and TRF Southern blot, flow-FISH, and TESLA using the same source DNA under similar laboratory conditions and we report the results of these comparisons throughout.

In reference to comparing data generated from different labs or different generations of flow cells, we have done both in our first rebuttal in comparing genomic DNA from HG002 cells sourced from the Coriell institute and sequenced in our laboratory and the Karlseder laboratory, reproduced below (and included in the supplement of the revised manuscript):

Manuscript	25th Percentile	Median	Mean	75th Percentile
Schmidt, et al. (Karlseder, TeloSeq)	4011	4979	5247	6115
Sanchez, et al. (Artandi, Telometer)	4209	5064	5307	6110

In this revision we have also made the addition of the chromosome, allele-specific telomere length distribution measurements made in HG002 cells using the diploid, allele-annotated HG002 genome as a reference genome. We have edited the text to reflect this addition, the methods section to explain the procedure for aligning to an allele-annotated diploid genome, and provided the allele-specific telomere measurements as raw data in the supplementary spreadsheet containing the aggregated telomere measurement data for all experiments in the manuscript. These chromosome-specific measurements are in strong agreement with those produced by the Karlseder lab in their manuscript and those presented for the HG002 cell line in Karimian et al *Science* 2024. Together, these data present overwhelming evidence that telomere measurement by long-read sequencing is highly reproducible even in independent laboratories using orthogonal library preparation and bioinformatic methods.

We have also demonstrated the reproducibility of telomere measurement using our method across different generations of ONT flow cells in Figure S8 (previously S7) of our manuscript, reproduced below:

Finally, with respect to the subtelomeric region remaining on digested genomic DNA being less than 500 bp in length, we believe that our experiments quantifying the extent of subtelomeric DNA remaining on DNA which has been subjected to restriction digestion provide empirical evidence that there can be significantly more subtelomeric DNA spared by restriction enzymes than is theoretically calculated by examining a reference genome and there are a variety of plausible explanations

for this fact: methylation, subtelomeric sequence variation, and the sensitivity of phosphor imaging being insufficient to quantify the distribution of subtelomeric sequence left on digested genomic DNA, particularly when compared to direct sequencing of DNA following restriction digestion.

5. *Comment: As the authors have indicated that the contradicting data remain un-resolved. Can the authors provide a reference to the claim "The linear regressions of the various telomere length summary statistics versus age in this cross-sectional cohort also demonstrate that the relationship between age and length of shorter telomeres is less steep than it is for longer telomeres."*

Response: The reference for this claim is Figure 3C, and the revised version of the manuscript was previously edited to make this point more clearly, see below:

"...Strikingly, the third quartile telomere length decreases more steeply with age than the first quartile (Fig. 3C)..."

Additionally, to bolster our observation regarding the fraction of telomeres of varying length in hESC TIN2 mutants, we re-sequenced telomeres from the same source sample as in the original data to obtain 1661 additional heterozygous mutant telomere measurements and 271 additional homozygous mutant telomere measurements. Figure 2 has been updated with the new data, but remains unchanged in structure or interpretation, so our original observations stand despite greatly increasing the depth of sequencing for these samples.

Comment: Please provide the reads depth for each chromosomal end.

Response: This data is available in our response to the first round of review. That said, due to concerns about chromosome end-mapping for samples without ideal reference genomes we have removed panel 2i from Figure 2 as it is not essential for the thrust of the argument presented in the manuscript.

6. *Comment: As commented above, the read depth may be the reason of such heterogeneity of telomere length elongation.*

Response: As we discussed in our response to previous comments, we performed additional sequencing of HEK293T + GFP cells to obtain higher read depth at each chromosome arm for this experiment and obtained significantly higher coverage across all chromosome arms. Based on the results of our bootstrapping analyses we do not believe additional read coverage would alter the interpretation of the results of this experiment. Panel 2i has been removed from figure 2, but the new figure contains the additional measurements made during the revision sequencing experiment.

7. *Comment: Here, I would like to quote the authors' comment from their rebuttal for point 6 "We do not believe that the difference in the 0-1000 bp fraction between the various healthy donor groups is significant and the difference that is being observed is likely due to the small population studied in this manuscript." Are these difference consistent and informative?*

Response: There is a consistent difference in the shortest telomere fractions between populations of cells with defective telomere maintenance machinery and those without and this is reflected in both the in vitro experimental data and in the cohorts of healthy and diseased individuals. As we discuss in the manuscript, we believe this suggests the telomere length distribution contains additional information which reflects the underlying integrity of the telomere maintenance machinery which was previously inaccessible with other methods that only measure the mean telomere length. While we do not and cannot provide an exhaustive description of how specific perturbations alter the telomere length distribution, we do provide in vitro and in vivo evidence using relevant genetic models with known phenotypes and believe this discovery is a valuable contribution to the literature. Future work should further elucidate how the telomere length distribution encodes interpretable information about the underlying state of telomere maintenance in the cell. For example, the manuscript co-submitted alongside ours (Schmidt, et al...and Karlseder) demonstrates that examining the coefficient of variance of the telomere length distribution can distinguish cells which utilize telomerase for telomere maintenance from those which utilize ALT.

8. *Comment: Based on the author's description and rebuttal to point 2, no enrichment of telomere-containing genomic DNA fragments was performed. Given the telomeric content of human diploid cells is only 0.015% of the total*

genome the vast human genome, an enrichment step seem to be necessary for telomere length measurement using long-read sequencing platform. How many reads per sample can be obtained? The method described seems to be inconsistent throughout. While the authors indicated that “up to a total of 1000 ng” was used prior to sequencing adaptor ligation here, they also indicated that “at least 3 ug of genomic DNA should be used” on their response for point 11. Can the authors be clear on the methodology?

Response: Figure 1C demonstrates that enrichment is obtained by preferential ligation of the ONT sequencing adapter to telomeric ends. We demonstrated in the previous rebuttal and in the original manuscript that there is a variable number of telomere measurements which can be made from a library but show examples where over 20,000 measurements are obtained per sample. Finally, we have edited the methods section on multiplexing telomere capture libraries to respond to the final point about the inconsistency of input DNA. In fact, there is no inconsistency but the text was indeed unclear. It is correct that we routinely use at least 3 µg of genomic DNA prior to telomere capture, but the total amount of DNA is reduced throughout the library prep process (particularly, following bead purification steps). After the seqTether is annealed, we consider the telomere capture process complete and the only step remaining is sequencing adapter ligation. It is these “post-capture”, barcoded libraries which are then quantified and pooled to a total DNA weight of ~1000 ng in a single test tube. The pooled, barcoded libraries are then ligated onto the ONT sequencing adapter without any modifications from the originally described method. The methods section has been edited to clarify the multiplex methodology, and the relevant excerpt is reproduced below:

“The bead solution was then pelleted on a magnet, washed twice with 80% ethanol, allowed to dry on the magnet for 3 minutes, and then eluted in 60 µL of ONT elution buffer for 15 minutes at 37°C. **The eluate contains the capture oligo ligated DNA, which will be referred to as the “post-capture” sequencing library.** The complete volume of post-capture library was then ligated onto the ONT sequencing adapter (AMII for R9, NA for R10 libraries) per ONT’s barcode ligation sequencing protocol (5 µL sequencing adapter mix, 20 µL 5X Quick T4 DNA Ligase Buffer, 10 µL Quick T4 DNA Ligase, 5 µL nuclease-free H₂O) at room temperature for one hour. The rest of the library preparation and sequencing protocol is performed as for whole-genome sequencing above. Technical replicate comparisons of R9 and R10 flow cell chemistries and basecallers are available in the supplementary information (Fig. S7 to S8).

To prepare multiplexed telomere capture sequencing samples, post-capture libraries containing up to 8 uniquely barcoded samples were pooled by quantifying each post-capture library with the Qubit BR dsDNA quantification assay and then combining equal molecular weights of each post-capture library up to a total of 1000 ng. Sequencing adapter ligation and subsequent preparation for flow cell loading is as previously described using the 1000 ng pool of post-capture libraries as input.”

9. *Comment: While it is true that the long-read sequencing platforms (ONT and PacBio) can achieve better accuracy than TRF, flow-FISH, STELA and TeSLA, I do believe the current requirement of genomic DNA input for long-read sequencing platforms is much higher than these aforementioned methods. PCR based methods such as STELA requires <100ng of genomic DNA.*

Response: The section the reviewer is referencing in this comment reads as follows:

“Telomere measurements made by long-read sequencing are **comparatively much richer than TRF, flow-FISH** while requiring **less input DNA than either existing technique.** **PCR-based assays such as STELA and TeSLA can be limited by polymerase processivity** and therefore reflect shorter telomeres without illuminating the entire distribution of telomeres in the sample (Fig. S6).”

The comment about less input DNA referenced TRF and flow-FISH, but is not inclusive of PCR-based assays, which are discussed in the second sentence in the context of a different limitation: polymerase processivity.

10. *Comment: Please include the telomere-containing reads number in the table. In addition, the newly provided excel table (file name: 462782_1_data_set_8811944_s11nf9.csv) lacks clear column descriptions, making it difficult to interpret the data accurately. For instance, while HG002 shows 10465 telomere reads with a coverage of 3.3, DC1 has only 134 telomere reads but with a higher coverage of 7.7.*

Response: HG002 is a telomere capture experiment while DC1 was measured using WGS. As is shown in Figure 1C, much more overall sequencing coverage is required in order to obtain a representative sampling of the telomere length distribution

when using WGS long-read sequencing. 3.3 and 7.7 refer to “genomic coverage”, which is calculated as follows: (Total Gb Sequenced) / (Size of Human Genome (~3 Gb))

11. *Comment: Please provide the read depth of telomere reads for each samples shown. This is important for the readers to understand whether these inconsistencies are due to read depth. While it is true the current ONT method provides better accuracy to telomere length measurement, such accuracy is depended on the read depth as the authors have shown in their bootstrapping analysis.*

Response: The supplemental spreadsheet provides this information for all sequencing runs in the manuscript.

Reviewer #2

Summary

Sanchez et al had addressed most of the points I raised. Most significantly, they had replaced their neural network model with a regression model. Additionally, they have made the presentation of their data much clearer. The following are points that I had previously raised, but are not fully addressed yet:

Major points

1. *As before, I still have significant concerns about the misassignment of the telomeric reads to the wrong chromosomal arms. As also highlighted by the authors, the reliability of mapping of true telomeric reads to the correct chromosome arm will also depend on the quality of the reference genome. Thus, in the absence of a complete reference genome of HEK293T, mis-mapping of telomeric reads between chromosomal arms is likely to occur.*

I would therefore suggest that the authors (i) provide further evidence to show that telomeric reads are mapped to the correct arms, (ii) only highlight the chromosomal arms that can be reliably assigned, or (iii) to remove this panel (Figure 2i) from this figure as it is not as pertinent to their story.

Response: We have removed the chromosome-specific measurement panel from figure 2 and edited the text to reflect this change.

2. *The statistics that I had requested should be quite easy for the authors to collect. However, only the number of telomeric reads were presented. It is unclear to me how many reads were generated in total. Again, these metrics would be very useful for readers who are interested in understanding the relative performance of different approaches for telomere enrichment.*

Response: We have included the total number of reads and % of telomeric reads as additional columns in the supplemental spreadsheet.

3. *These numbers should be presented as part of the table for point 7a. Here, it is also quite surprising to me that most of the reads were observed from the C strand while in the reply to question 3 by Reviewer #3, the authors had observed more reads with the AGGGTT motif (i.e. G strand) than with the AACCCCT motif (i.e. C strand). The two pieces of data that the authors had presented therefore seem to contradict each other?*

Response: These are not in contradiction, but the apparent contradiction is an artefact of how these data are generated. The analysis of telomeric motifs was done by identifying telomeric motif sequences from alignment (.bam) files. Haploid genome assemblies (like CHM13v2.0 / hs1) represent the chromosomal P-arm exclusively with the C-strand sequence, and the Q-arm exclusively with the G-strand sequence. Thus, following alignment, any telomere read aligning to the P-arm will contain the C-strand sequence and vice versa. In order to determine the true strand identity of the alignment one must know the Watson-Crick strand corresponding to that read (+ or -) and the direction of the read (forward or reverse), and both of this information is contained in the read metadata within the .bam alignment file. The telometer script pulls both of these pieces of information from the .bam and saves it in the output by default and I used all of this information together to identify if the C or G-strand of the telomere was sequenced. That said, when counting the sequence of telomeric motifs, the sequence is used as it is stored in the .bam file, resulting in roughly equal representation of TTAGGG or CCCTAA motifs.

- 4. The authors had stated in their revised text that: "including more rapid shortening of the third quartile relative to the first (but this trend does not reach statistical significance)." If the result does not reach statistical significance, there is insufficient evidence for the authors to conclude that the third quartile is shortening more rapidly than the first. I would therefore encourage the authors to refrain from making such a claim.*

Response: We have removed this claim from the text.

Minor

- 1. If the comparison is between the author's approach with that from citation 26, this should be made very clear. The authors had reported that this difference is due to the sequencing platform in the legend of Figure S3. However, these differences could instead have occurred due to how these libraries were prepared rather than due to the sequencing platforms.*

Response: The text has been edited to further highlight that the PacBio data was sourced from the publication cited in 26.

Reviewer #3 (comments to the authors)

Overall the paper is very improved, and they have also addressed many of the questions about k-mer differences and mapping.

Response: Thank you for your support.

However, a request for the code and methods for the classifier was not completed, as far as I can tell? It just says "We will make this available for review during this resubmission."

Response: The links to the Telometer repository was included in the methods section of the resubmission manuscript. The classifier code and pre-trained models are on a separate github repository: https://github.com/santiago-es/SES_NatComms24

The Telometer code is available via github at: <https://github.com/santiago-es/Telometer>